# Single-cell RNA-sequencing reveals distinct patterns of cell state heterogeneity in mouse models of breast cancer

Syn Kok Yeo[1†]*, Xiaoting Zhu[2,3†], Takako Okamoto[1], Mingang Hao[1], Cailian Wang[4], Peixin Lu[2,4], Long Jason Lu[2,3]*, Jun-Lin Guan[1]*

[1]Department of Cancer Biology, University of Cincinnati College of Medicine, Cincinnati, United States; [2]Division of Biomedical Informatics, Cincinnati Children's Hospital Research Foundation, Cincinnati, United States; [3]Department of Electrical Engineering and Computer Science, University of Cincinnati College of Engineering and Applied Science, Cincinnati, United States; [4]School of Information Management, Wuhan University, Wuhan, China

**Abstract** Breast cancer stem cells (BCSCs) contribute to intra-tumoral heterogeneity and therapeutic resistance. However, the binary concept of universal BCSCs co-existing with bulk tumor cells is over-simplified. Through single-cell RNA-sequencing, we found that Neu, PyMT and BRCA1-null mammary tumors each corresponded to a spectrum of minimally overlapping cell differentiation states without a universal BCSC population. Instead, our analyses revealed that these tumors contained distinct lineage-specific tumor propagating cells (TPCs) and this is reflective of the self-sustaining capabilities of lineage-specific stem/progenitor cells in the mammary epithelial hierarchy. By understanding the respective tumor hierarchies, we were able to identify CD14 as a TPC marker in the Neu tumor. Additionally, single-cell breast cancer subtype stratification revealed the co-existence of multiple breast cancer subtypes within tumors. Collectively, our findings emphasize the need to account for lineage-specific TPCs and the hierarchical composition within breast tumors, as these heterogenous sub-populations can have differential therapeutic susceptibilities.

*For correspondence:
yeosn@ucmail.uc.edu (SKY);
bioinfo@gmail.com (LJL);
guanjl@ucmail.uc.edu (J-LG)

†These authors contributed equally to this work

Competing interests: The authors declare that no competing interests exist.

## Introduction

The mammary gland is a bi-layered epithelial organ which consists of basal as well as myoepithelial cells in the outer layer. The inner epithelium comprises of luminal cells which can be broadly classified into the hormone-sensing lineage (estrogen receptor + or progesterone receptor +, ER+/PR+) and the secretory alveolar lineage (*Bach et al., 2017*). As with most other organs, a hierarchical relationship can be established between the differentiation states of the distinct cell lineages within the mammary gland (*Visvader and Clevers, 2016*). During embryonic developmental stages, bipotent mammary stem cells (MaSCs) with hybrid basal-luminal gene expression patterns and the ability to give rise to both layers of the developed mammary epithelium have been described (*Pal et al., 2017*; *Spike et al., 2012*; *Wuidart et al., 2018*). Postnatally, the hybrid basal-luminal gene signatures diminish and eventually leads to the bifurcation of unipotent basal and luminal lineage stem/progenitor cells respectively by the onset of puberty in mice (*Giraddi et al., 2018*; *Pal et al., 2017*; *Van Keymeulen et al., 2011*; *Wuidart et al., 2018*). Recent insights from lineage tracing in mouse experiments have also revealed that the ER+ and ER- luminal lineages can respectively self-sustain, through multiple cycles of pregnancy, lactation and involution (*Van Keymeulen et al., 2017*;

*Wang et al., 2017*). Altogether, there is cumulative evidence to support the existence of primitive bipotent fetal-MaSCs (fMaSCs) (*Spike et al., 2012*; *Wuidart et al., 2018*), quiescent adult-MaSCs (aMaSCs) (*Fu et al., 2017*; *Rios et al., 2014*), basal-lineage SCs (*Shackleton et al., 2006*; *Van Keymeulen et al., 2011*), ER⁻ luminal SCs (*Wang et al., 2017*) and ER⁺ luminal SCs (*Van Keymeulen et al., 2017*; *Wang et al., 2017*) at varying timepoints of mammary gland development in the mouse. As such, there is a milieu of long-lived SCs/progenitors within the mouse mammary gland that could be candidates for transformation and potential tumor-initiating cells, but each with properties that are unique to a particular lineage or developmental timepoint. Alternatively, neoplastic transformation of cells can also lead to plasticity (*Koren et al., 2015*; *Molyneux et al., 2010*) and the acquisition of any one of the aforementioned stem/progenitor-like states could have a unique role in tumor progression.

In the context of cancer, it has been proposed that differentiation hierarchies also exist among tumor cells, with cancer stem cells (CSCs) residing at the apex of the hierarchy within tumors (*Nguyen et al., 2012*). Although much progress has been made since the initial prospective isolation of CSCs in breast tumors (*Al-Hajj et al., 2003*), major gaps remain in the conceptual framework of breast CSCs (BCSCs), particularly in light of recent insights from the normal mammary differentiation hierarchy (*Fu et al., 2017*; *Nguyen et al., 2018*; *Van Keymeulen et al., 2017*; *Van Keymeulen et al., 2011*; *Wang et al., 2017*; *Wuidart et al., 2018*). An additional confounding factor has been the description of multiple distinct markers that can enrich for BCSC populations such as CD44⁺/CD24⁻ (*Al-Hajj et al., 2003*), ALDH⁺ (*Ginestier et al., 2007*), HER2⁺ (*Ithimakin et al., 2013*), CD29^hi (*Kouros-Mehr et al., 2008*) and side population cells (*Britton et al., 2011*). Recent studies have shown that the CD44⁺/CD24⁻ and ALDH⁺ markers identify minimally overlapping populations and each of these sub-populations exhibit epithelial to mesenchymal (EMT)-like and mesenchymal to epithelial (MET)-like characteristics, respectively (*Liu et al., 2014*). Distinct BCSC populations have also been shown to have differential preferences for signaling and metabolic pathways (*Luo et al., 2018*; *Roarty et al., 2017*; *Yeo et al., 2016*), suggesting the need to account for these BCSC populations separately when it comes to therapeutic targeting. It is also important to note that not all previously characterized BCSC markers have been investigated concurrently in an exhaustive manner (*Hwang-Verslues et al., 2009*). Accordingly, the possibility that more non-overlapping markers that enrich for other distinct BCSC populations remain to be determined. Since some of the previously described BCSCs that have increased tumor propagating potential may exist in a progenitor or EMT state that does not strictly correspond to mammary stem cells, we will utilize the term tumor propagating cells (TPCs) when referring to cells with increased tumorigenic potential (*Giraddi et al., 2018*).

The mammary differentiation hierarchy has also been pivotal in guiding the stratification of breast cancer subtypes (*Visvader and Clevers, 2016*). Immunohistochemical analysis of estrogen receptor (ER), progesterone receptor (PR) and HER2 receptor enables a crude association with the hormone-sensing lineage (ER⁺, PR⁺) and basal-like (ER⁻, PR⁻, HER2⁻) subtypes of breast cancer. Parallels between the intrinsic molecular subtypes of breast cancer (*Perou et al., 2000*; *Prat et al., 2010*) and mammary differentiation states can also be observed with basal-like breast cancers being associated with mouse fMaSCs (*Pal et al., 2017*; *Spike et al., 2012*), claudin-low breast cancers with mouse aMaSCs (*Fu et al., 2017*) and luminal-A breast cancers with the hormone-sensing lineage. By extension, TPCs which occupy a less-differentiated cell state could correspond to a different breast cancer subtype relative to bulk tumor cells. This raises the question of whether multiple breast cancer subtypes co-exist within a tumor (*Yeo and Guan, 2017*). An increasing number of observations support this notion (*Chung et al., 2017*; *Roarty et al., 2017*) but breast cancer subtype heterogeneity within mouse models of breast cancer has not been well characterized.

In this study, using molecular profiling and single-cell RNA sequencing (scRNA-seq) of multiple mouse models of breast cancer, we demonstrated that various tumors driven by different oncogenic events contained tumor cells spanning distinct spectra of cell states within the mammary epithelial hierarchy. We further showed that each of these mammary tumors contained putative lineage-specific TPCs, rather than a universal BCSC population. Accordingly, we utilized CD14 to isolate alveolar progenitor-like TPCs from Neu tumors based on the newly revealed hierarchy of cell states within the tumors. Importantly, the hierarchical heterogeneity that was observed in these tumors was accompanied by the co-existence of cells associated with multiple PAM50 breast cancer subtypes within each of the tumors examined.

## Results

### Tumor cells from different mouse models of breast cancer cluster separately with distinct signatures of mammary lineage markers

To gain insights into the heterogeneity within mammary tumors, we performed single-cell transcriptional profiling of tumor cells from multiple mouse models representing both luminal and basal-like subtypes of breast cancer (*Figure 1A*). Both MMTV-Neu (*Guy et al., 1992b*) and MMTV-PyMT (*Guy et al., 1992a*) driven tumors (designated as Neu and PyMT tumors, respectively) have been well-characterized and used extensively to illuminate the roles of various oncogenic and tumor suppressive pathways that are relevant in Her2$^+$ and other luminal breast cancers. Essentially, the MMTV-Neu tumors examined were ER$^-$/PR$^-$/HER2$^+$ while the MMTV-PyMT tumors were ER$^-$/PR$^-$/HER2$^{lo}$ (*Figure 1—figure supplement 1A*). Complementarily, *Brca1$^{F/F}$ Trp53$^{F/F}$ Krt14-Cre* mice develop mammary tumors (*Liu et al., 2007*) (designated as BRCA1-null tumors) which mimic basal-like breast cancers and were ER$^-$/PR$^-$/HER$^-$ (*Figure 1—figure supplement 1A*). In these experiments, PyMT, Neu and BRCA1-null tumors were derived and extracted from congenic FvB background mice when tumors were approximately 1000 mm$^3$. Tumors were dissociated into single-cell suspensions before sorting for epithelial CD24$^+$ Lin$^-$ (CD31$^-$ CD45$^-$ Ter119$^-$) cells (*Figure 1A*, *Figure 1—figure supplement 1B–C*), to enrich for tumor cells and reduce stromal cell contamination. Consequently, isolated tumor cells from the three different tumor types were subjected independently to the Chromium 10x droplet-based single-cell RNA-sequencing (sc-RNAseq) platform, before pooling together cDNA libraries from all tumors for sequencing. Biological replicates for each tumor type were sequenced independently in a second batch. A total of 11842 cells were sequenced from all three tumor types and after rigorous quality control filtering (*Figure 1—figure supplement 2A–D*), 9983 cells (4154, 3545 and 2284 cells for Neu, PyMT and BRCA1-null tumors, respectively) were retained for subsequent analyses. On average, between 2205 to 3363 unique genes were detected for cells from each tumor sample (*Figure 1—figure supplement 2E*). Despite sequencing the tumor replicates in a separate batch, initial analysis of cells through dimensionality reduction algorithms (Bioconductor: scran) revealed that cells from both replicates were overlapping to a large degree (*Figure 1—figure supplement 3A*). Moreover, replicates of respective tumor types were clustering together (*Figure 1—figure supplement 3B*) and had a high degree of concordance (Pearson correlation coefficients > 0.97 between respective replicates for all tumor types, *Figure 1—figure supplement 3C–E*), indicating that batch effects were minimal. Among these cells, the majority were more closely associated with a G1 cell cycle state and less than 2.5% of cells were either in G2/M or S phases (*Figure 1—figure supplement 4A–D*). Additionally, normal mammary epithelial cell (MEC) contamination was inferred to be minimal among the cells examined (Cells without CNV in Neu: 0.4%, PyMT: 0%, BRCA1-null: 8.8%) (*Figure 1—figure supplement 4E–F*).

Initial visualization of the data through t-distributed stochastic neighbor embedding (t-SNE) plots showed that Neu, PyMT and BRCA1-null tumor cells formed three distinct groups (*Figure 1B*) with minimal overlap between cells of distinct tumor types, which is consistent with each of these models being driven by distinct oncogenic events. These three main groups could be further subdivided into six clusters upon unsupervised clustering analysis by a shared nearest-neighbor clustering approach (*Figure 1C*). To broadly characterize the different clusters, a gene expression heatmap comprising the top-ranked differentially upregulated genes between each cluster was generated (*Figure 1D*). The basal-like BRCA1-null cells had increased expression of basal-associated genes such as *Krt14, Vim* and *Sparc* (*Figure 1D–E*). Intriguingly, the BRCA1-null tumor cells could be segregated further into cells with basal (Cluster 4) and mesenchymal (Cluster 5) features, respectively (*Figure 1D*). Cluster 4 cells expressed higher levels of epithelial genes (*Cldn4, Cldn3, Epcam*) whereas Cluster 5 was defined by expression of mesenchymal genes (*Vim, Sparc, Col3a1, Bgn*) (*Figure 1D*). A very small proportion of PyMT cells can also be found within both of these clusters and the presence of basal cells (KRT14$^+$ leader cells or CD29$^{hi}$ cells) have been described in this tumor model (*Cheung et al., 2013*; *Yeo et al., 2016*). In the case of PyMT tumor cells, higher levels of *Ltf, Spp1, Anxa1* and *Cldn4* distinguish them from Neu and BRCA1-null tumor cells (*Figure 1D and F*), and some of these genes have been associated with luminal progenitors (*Nguyen et al., 2018*). In addition, levels of the luminal progenitor marker *Aldh1a3* was exclusively detected in PyMT tumor cells (*Figure 1F*), indicating that these tumors consist of cells associated with a primitive luminal cell state (*Eirew et al., 2012*). Neu tumor cells can be divided into Cluster 1 and Cluster 2,

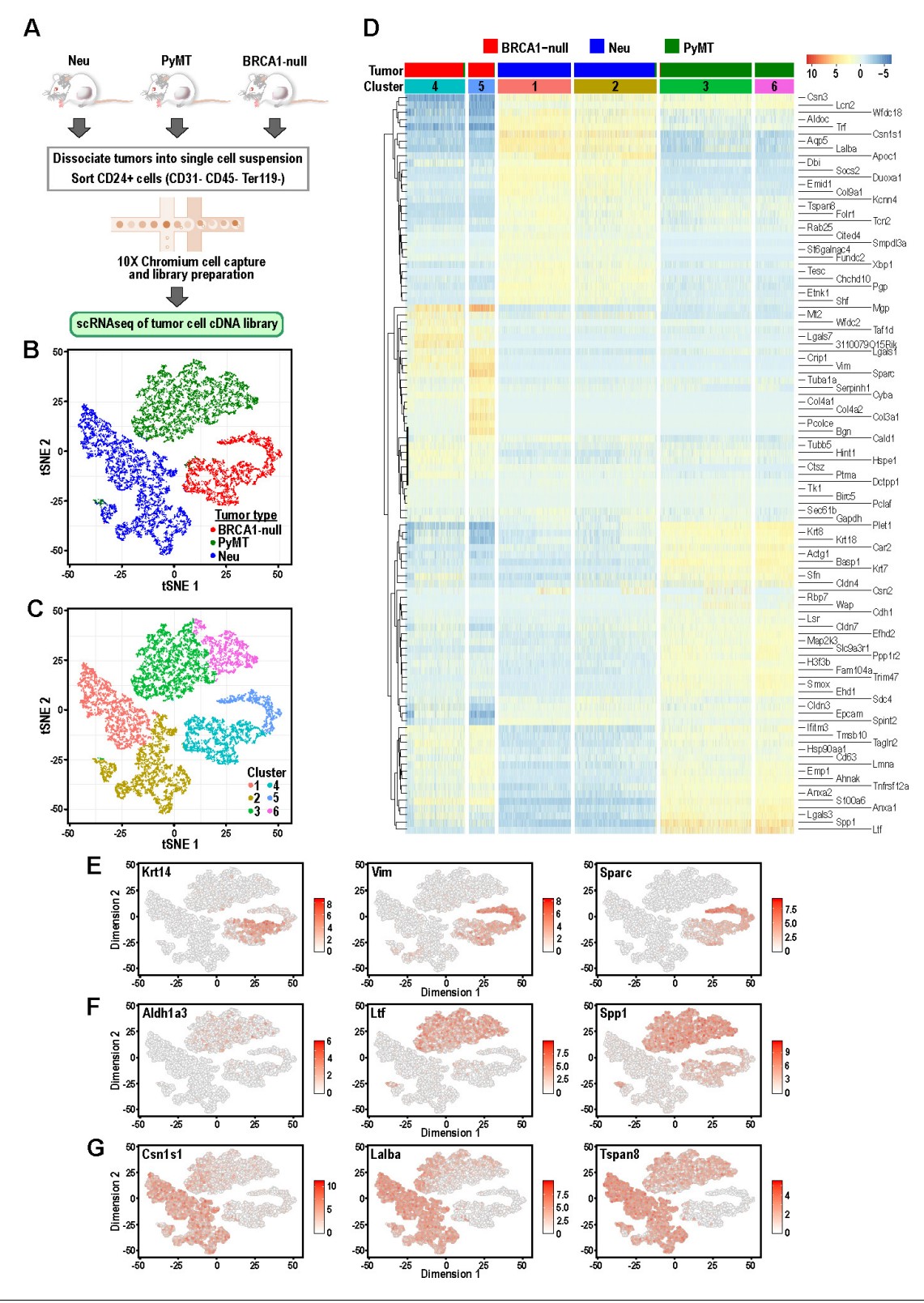

**Figure 1.** Tumor cells from distinct mouse models of breast cancer cluster separately and can be differentiated by the expression of mammary lineage markers. (A) Schematic of workflow for the isolation of mammary tumor cells from BRCA1-null, PyMT and Neu tumors for scRNA-seq. (B) t-SNE plots of mammary tumor cells colored by tumor sample; Neu (blue), PyMT (green), BRCA1-null (red). (C) t-SNE plots of mammary tumor cells colored by clusters

*Figure 1 continued on next page*

eLife Research article

Cancer Biology | Genetics and Genomics

*Figure 1 continued*

1–6. (D) Heatmap of top differentially upregulated genes between tumor clusters. (E–G) t-SNE plots of tumor clusters colored by the normalized log-transformed expression of the indicated genes classified as (E) basal genes, (F) luminal progenitor associated genes and (G) alveolar lineage genes. The online version of this article includes the following figure supplement(s) for figure 1:

**Figure supplement 1.** Receptor status and representative gating strategy for sorting of Lin⁻ CD24⁺ mammary tumor cells from the distinct tumors.
**Figure supplement 2.** Quality control filtering of scRNA-seq prior to processing.
**Figure supplement 3.** Replicates of tumor samples exhibit a high degree of concordance and reproducibility.
**Figure supplement 4.** Cell cycle and copy number variation (CNV) prediction for individual tumor cells.

both expressing higher levels of genes associated with secretory alveolar differentiation such as *Csn1s1, Lalba* and *Tspan8* (*Figure 1D and G*). Levels of some alveolar differentiation genes such as *Cited4* and *Tspan8* were marginally higher in Cluster 1, indicating that Cluster 1 cells were more differentiated along the alveolar lineage relative to Cluster 2 (*Figure 1D*). Altogether, these results revealed that mammary tumors driven by different oncogenic events contained clusters of tumor cells that were associated with distinct developmental cell states with minimal overlap between the tumor types.

## Mammary tumors exhibit distinct patterns of hierarchical heterogeneity

To gain a better understanding of the distinct cell states that were represented by the mammary tumor models, we examined the single-cell transcriptomes of the tumor cells in comparison to that of normal mouse mammary epithelial cells (MECs) across unique developmental stages (nulliparous, mid-gestation, lactation and post-involution) (*Bach et al., 2017*). In the previous study, 20 cell states were identified and their putative identities determined (C1-C15: epithelial cell clusters, C16-C20: other cell types). The broad coverage of this normal MEC dataset across varying developmental stages would allow for the prospective identification of developmental processes hijacked by tumor cells in these mouse models (e.g. pregnancy mimicry, involution mimicry). The relative differences in gene expression pattern between 12000 cells (11637 cells after quality control) randomly selected from Bach et al.'s dataset (*Bach et al., 2017*) together with 9983 cells in our dataset were analyzed using Seurat v3 (*Figure 2A*). The Seurat v3 R package allows for integration of separate experimental datasets across species and even varying experimental platforms (*Stuart et al., 2019*). We found that Neu, PyMT and BRCA1-null tumor cell clusters overlapped with distinct normal MEC clusters through this analysis (*Figure 2B*).

BRCA1-null tumor cells were distributed amongst distinct clusters associated with C13, basal cells during gestation (C13:Bsl-G) and C12, basal cells (C12:Bsl) (*Figure 2C*), consistent with our initial observations of these cells having increased expression of basal genes (*Figure 1D–E*). The expression of *Col4a1* is relatively higher in C13:Bsl-G cells (*Figure 2D*), differentiating them from C12:Bsl associated tumor cells. In addition, BRCA1-null cells also clustered amongst C10, alveolar progenitors during gestation (C10:AvP-G), C15, PROCR+ cells that arise during lactation (C15: Prc) and C18, fibroblasts with mesenchymal features (C18:Fibroblasts) (*Figure 2C*). The expression of *Col1a2* was evidently higher in C18:Fibroblasts (*Figure 2E*), which is typical of their extracellular matrix deposition functions and fits with our earlier observation of cluster 5 (*Figure 1D*), which exhibit increased mesenchymal gene expression. It is also worth noting that a spectrum of BRCA1-null tumor cells can be found distributed between clusters C12:Bsl and C10:AvP-G (*Figure 2C*). These cells could represent a population with hybrid basal-alveolar gene expression, which have also been documented by Bach et al. and were designated as C20 (*Bach et al., 2017*). This highlights the extensive degree of cell state heterogeneity within BRCA1-null tumors.

As for PyMT tumor cells (*Figure 2F*), they primarily clustered between C6, luminal progenitors from nulliparous glands (C6:LP-NP) and C7, luminal progenitors during involution (C7:LP-PI), consistent with their association with a luminal progenitor gene expression signature seen in our initial analysis (*Figure 1F*). A small proportion of PyMT cells were also converging with C10, alveolar progenitors during gestation (C10:AvP-G), indicating the priming of these luminal progenitor-like cells towards the alveolar lineage (*Figure 2F*). Interestingly, PyMT tumor cells that were proximal to the C6:LP-NP cluster exhibited elevated *Aldh1a3* expression (*Figure 2G*), whereas there was an

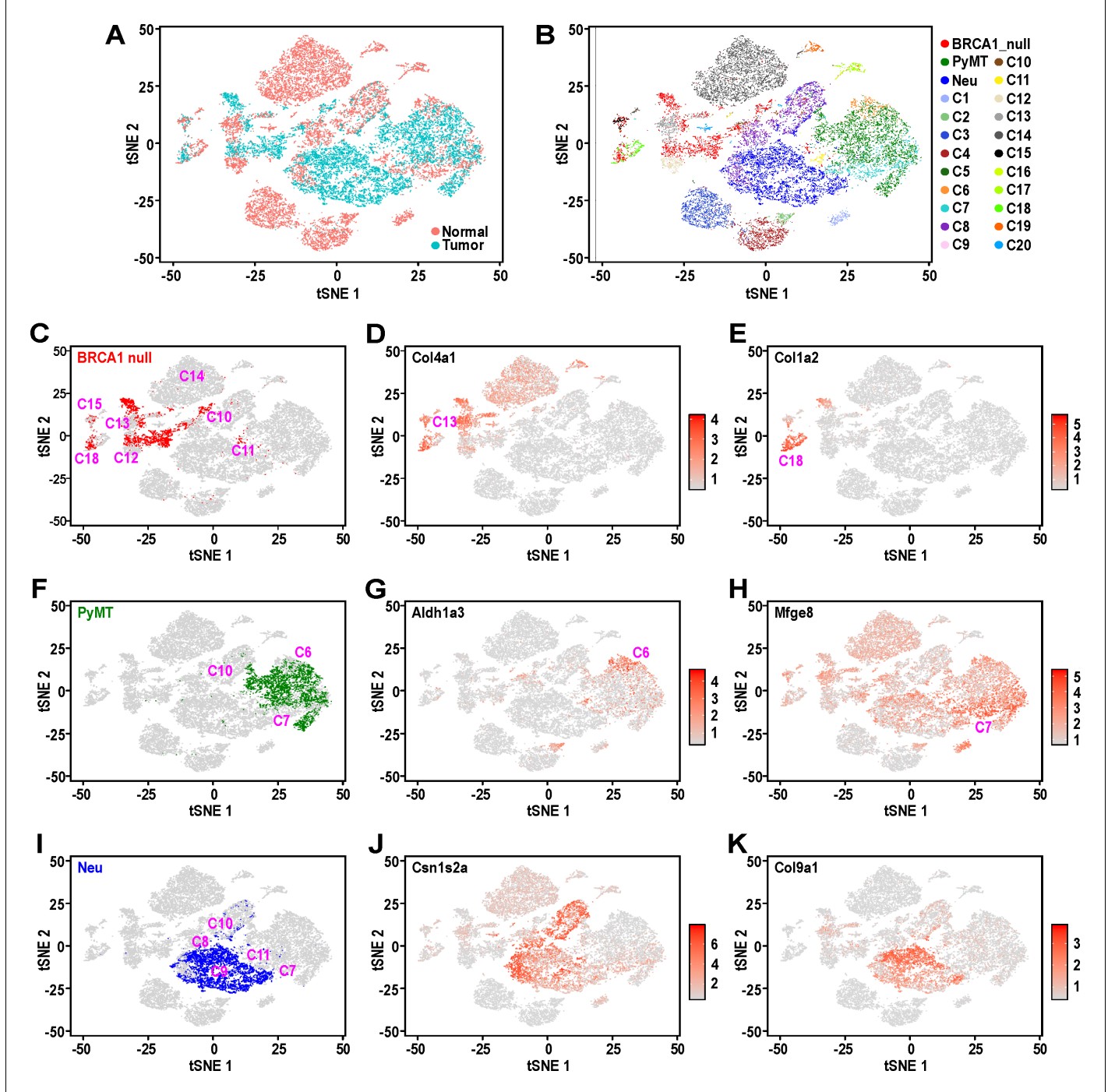

**Figure 2.** Mouse mammary tumors display distinct patterns of hierarchical heterogeneity. (**A**) t-SNE plots of mammary tumor cells (light blue) along with 12000 randomly selected normal MECs (pink) from Bach et al.'s dataset (*Bach et al., 2017*). (**B–I**) Similar t-SNE plots colored by (**B**) tumor sample and putative normal MEC identity (as described in *Bach et al., 2017*), (**C**) BRCA1-null tumor cells (red) with surrounding clusters, (**D**) normalized log-transformed expression levels of *Col4a1*, (**E**) normalized log-transformed expression levels of *Col1a2*, (**F**) PyMT tumor cells (green) with surrounding tumor clusters, (**G**) normalized log-transformed expression levels of *Aldh1a3*, (**H**) normalized log-transformed expression levels of *Mfge8*, (**I**) Neu tumor cells (blue) with surrounding clusters, (**J**) normalized log-transformed expression levels of *Csn1s2a* and (**K**) normalized log-transformed expression levels of *Col9a1*.

The online version of this article includes the following figure supplement(s) for figure 2:

**Figure supplement 1.** Independent iterations for Seurat v3 analysis and Monocle trajectory analysis of mammary tumor cells.

enrichment of cells with high *Mfge8* expression towards the C7:LP-PI cluster (*Figure 2H*), suggesting that the post-involution state is associated with increased *Mfge8* expression.

Comparison of Neu tumor cells with the various clusters in Bach et al.'s dataset showed that they were mostly clustered between C9, alveolar differentiated cells during lactation (C9: AvD-L) and C11, alveolar progenitor cells of lactating glands (C11:AvP-L)(*Figure 2I*). A portion of tumor cells adjacent to C11 also extend into and overlap with C7, luminal progenitor cells from post-involution (C7:LP-PI). We also noted Neu tumor cells scattered between C8, alveolar differentiated cells during gestation (C8:AvD-G) and C10, alveolar progenitor cells during gestation (C10:AvP-G). These results were consistent with the initial analyses for Neu tumor cells that exhibited a secretory alveolar gene expression signature (*Figure 1G*), and also suggested these Neu tumors contained cells which corresponded to a spectrum of differentiation states within the secretory alveolar lineage (i.e. C7:LP-PI → C11 and C10: AvP → C9 and C8: AvD). The increasing level of differentiation in this cascade among the population of Neu tumor cells was also supported by the gradual change in expression of the milk protein gene *Csn1s2a* in the main cluster of Neu tumor cells (*Figure 2J*). Additionally, the distinct pattern of *Col9a1* expression among Neu tumor cells (*Figure 2K*) suggested the clustering of alveolar cells from gestation and lactation timepoints respectively. Overall, this analysis utilizing Seurat v3 (*Figure 2*) was robust and stable, as evidenced by the same pattern of overlap between tumor and normal cells from two other independent iterations, utilizing independent sampling of random cells from Bach et al.'s dataset (*Figure 2—figure supplement 1A*).

In order to provide a clearer definition of cells with similar gene expression patterns, standard graph-based clustering approach was applied on the combined dataset of tumor and normal cells and this integrated analysis resulted in 17 clusters (Cluster 0–16) (*Figure 3A*). Tumor cells were clustered with certain normal cells through this analysis, indicating shared gene expression patterns between tumor cells and certain cell states (*Figure 3B*). Based on the heatmap of genes defining these clusters (*Figure 3C*), it was apparent that the clustering was influenced largely by genes that define mammary cell lineages or states such as *Spp1, Ltf, Acta2, Lalba, Wap, Prlr, Areg* and *Col1a2*. Accordingly, the number of tumor cells that were assigned into clusters with normal cells of a particular cell state could be determined in detail (*Figure 3D*). The majority of BRCA1-null cells were assigned into clusters 5 and 8 (41.5% and 20.6% of total BRCA1-null cells respectively), which contained C12:Bsl, C13:Bsl-G and C20 cells. The other cell states that were associated with more than 1% of BRCA1-null cells were LP, AvP, AvD, C14:Myoepithelial, C15:Prc and C18:Fibroblasts. In the case of PyMT tumor cells, 93% of cells were assigned into cluster 0, which corresponded to the luminal progenitor populations C6:LP and C7:LP-PI (*Figure 3D*). Most of the remaining PyMT tumor cells were assigned into clusters 3 and 4, that were associated with C7:LP-PI, AvP and AvD cells. Neu cells were also assigned into Clusters 3 and 4 (33% and 7.7% of total Neu cells respectively), further affirming the alveolar lineage association of these tumor cells (*Figure 3D*). The bulk of Neu tumor cells (58%) were assigned into cluster 2, corresponding to C8:AvD-G cells. Overall, the association between tumor and normal cells through clustering (*Figure 3*) corroborated the observations from t-SNE plots (*Figure 2*), alleviating complications that may arise when interpreting distances in low dimensional space (*Figure 2*, t-SNE). Taken together, our comparative analyses of various tumor cells superimposed with a recently described single-cell transcriptome dataset of normal MECs revealed the unique segments of the mammary differentiation spectrum occupied by each tumor type, with potentially distinct lineage-specific TPCs in each of the tumors examined.

As an orthogonal approach, we utilized Monocle pseudotime analysis, which allows the visualization of gradual but stochastic transitions between cells (*Trapnell et al., 2014*) to examine the relative relationship between tumor cells from different models across a continuum of differentiation states. The computed cell state trajectory for these tumor cells can be represented by ascending diffusion pseudo-times (DPT) that span from 0 to 43 DPT (*Figure 2—figure supplement 1B*). Evidently, a main root and two branches can be observed, but only a small proportion of tumor cells from distinct tumor types overlapped at the intersection of these branches (*Figure 2—figure supplement 1B–C*). Accordingly, BRCA1-null tumor cells exhibited the lowest DPT, PyMT tumor cells had moderate DPT, whereas Neu tumor cells were associated with the highest DPT (*Figure 2—figure supplement 1D–E*). Together, these results further substantiated the fact that these different tumor models comprised of cells which corresponded mostly to distinct spectrums of cell states.

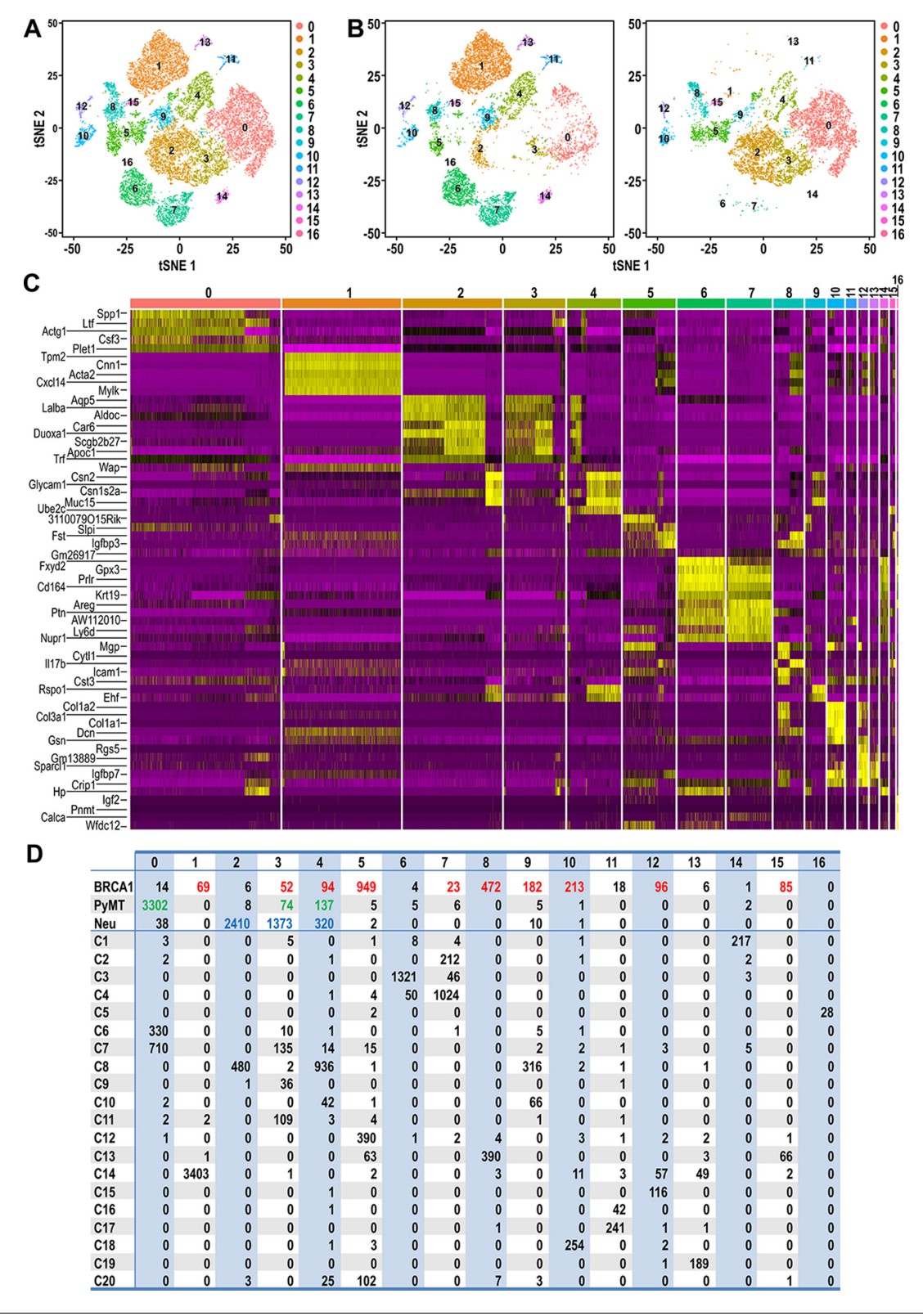

**Figure 3.** Clustering of tumor and normal datasets reveal distinct spectrums of cell states within each tumor. (A) t-SNE plots of tumor and normal datasets colored by clusters 0–16. (B) t-SNE plots showing the composition of normal and tumor cells respectively in clusters 0–16. (C) Heatmap of top upregulated genes defining cluster 0–16. (D) Table showing the number of tumor and/or normal cells within clusters 0–16. Tumor cell numbers which were more than 1% of the total cells for each respective tumor type are colored; BRCA1-null (red), PyMT (green) and Neu (blue).

## Analysis of unique patterns of intra-tumoral heterogeneity in different mouse models of breast cancer

Intra-tumoral heterogeneity is increasingly recognized as a critical factor for cancer metastasis, relapse and drug resistance (*Tabassum and Polyak, 2015*). Therefore, we focused on dissecting the single cell transcriptomes of tumor cells within individual models of mammary tumors. Both replicates were included for each tumor model and the distribution of cells was even amongst the clusters for all tumor types (*Figure 4—figure supplement 1A–C*). The BRCA1-null tumors can be clustered into five main clusters, designated as BRCA1-1 to BRCA1-5 (*Figure 4A–B*). The BRCA1-1 cluster consisted of proliferating cells with increased expression of cell cycle related genes such as *Birc5, Tyms* and *Mki67*. BRCA1-5 was a cluster with strongest expression of prototypical basal genes such as *Krt14* and *Igfbp5*. Interestingly, BRCA1-2 cells exhibited a partial EMT gene expression pattern with moderate levels of collagens and the highest levels of EMT transcription factors such as *Yap1, Twist1* and *Zeb1* (*Figure 4B*). On the other hand, BRCA1-3 cells expressed the highest levels of genes associated with fibroblasts (*Figure 4B*) and this would fit with our previous observations of a sub-population of BRCA1 null cells with mesenchymal characteristics (*Figures 1–3*). Within this predominantly basal tumor type, cells with features of AvPs (BRCA1-4) were also present with elevated levels of *Cldn3, Malat1, Krt18, Wfdc18* and *Mfge8*. Once again, the extensive heterogeneity within these BRCA1-null tumors were apparent along with the genes and pathways for each of these clusters (*Figure 4—source datas 1–2*).

For PyMT cells, we identified three clusters, PyMT-1 to PyMT-3, which corresponded to LP-Involution (LP-PI) cells, Av-primed LPs and LPs respectively (*Figure 4C–D*). The involution genes elevated in PyMT-1 included *Mfge8, Csf3, Spp1* and *Ltf* (*Figure 4D*, *Figure 4—source data 3*) which corresponded well to enrichment of gene ontologies (GO) such as 'lipopolysaccharide-mediated signaling' and 'positive regulation of phagocytosis' (*Figure 4—source data 4*). Apart from expressing higher levels of alveolar differentiation genes (*Lalba, Wap, Csn1s1*), PyMT-2 cells were also found to express higher levels of genes categorized under the GO, 'oxidative phosphorylation' (*Figure 4—source data 4*). Contrastingly, upregulated genes for PyMT-3 cells were enriched for those categorized under 'glycolytic process' (*Figure 4—source data 4*), highlighting the possibility of metabolic heterogeneity between these PyMT tumor populations.

Additionally, we also performed scRNA-seq analysis on a 4T1 transplant tumor but this sample was not included in prior analysis (*Figures 1–3*), to avoid confounding factors such as genetic background. Nonetheless, intra-tumoral analysis revealed the presence of three clusters designated as 4T1-1 to 4T1-3 (*Figure 4—figure supplement 2A*). Cluster 4T1-1 showed increased co-expression of IFN signaling genes (*Irf1, Irf8, Cxcl9, Cxcl10*) and the proliferation marker gene *Mki67* (*Figure 4—figure supplement 2B*). Immunohistochemical staining for IRF1 and Ki67 in serial sections of 4T1 tumors validated the presence of IRF1[+] and Ki67[+] cells within the same region of these tumors (*Figure 4—figure supplement 2C*), suggesting that activation of interferon signaling could be a driver for tumor growth in this model. Another noteworthy feature of cells within this model was the increased expression of MHC-II molecules such as *H2-Ab1* and *H2-Aa* in cluster 4T1-3 (*Figure 4—figure supplement 2B*, *Figure 4—source data 5*). Although the expression of MHC-II have been generally attributed to be a feature of immune cells, tumor cell expression have been documented (*Forero et al., 2016*) and the significance of its increased expression in this sub-population of cells would be an interesting avenue for future studies.

Together, these studies illuminated the complex patterns of intra-tumoral heterogeneity for each model. They also provided interesting information for each cluster of cells within individual tumors, although the TPCs and putative markers to be used for isolation could not be deduced easily based on the current analysis so far, except for that in Neu cells discussed below.

## Identification of CD14[hi] cells with increased tumorigenic potential in MMTV-Neu tumors

Examination of Neu tumor cells revealed four clusters (Neu-1 to Neu-4) with well-defined characteristics to allow illustration of their hierarchical relationships (*Figure 4E and F*). Neu-3 cells had increased expression of milk production related genes such as *Lalba* and can be defined by higher expression of *Cited4, Cd36* and *Aqp5* which corresponded to the AvD cell state *Figure 4F*, *Figure 4—source datas 6–7*). Although cells within the Neu-2 cluster also exhibited increased milk

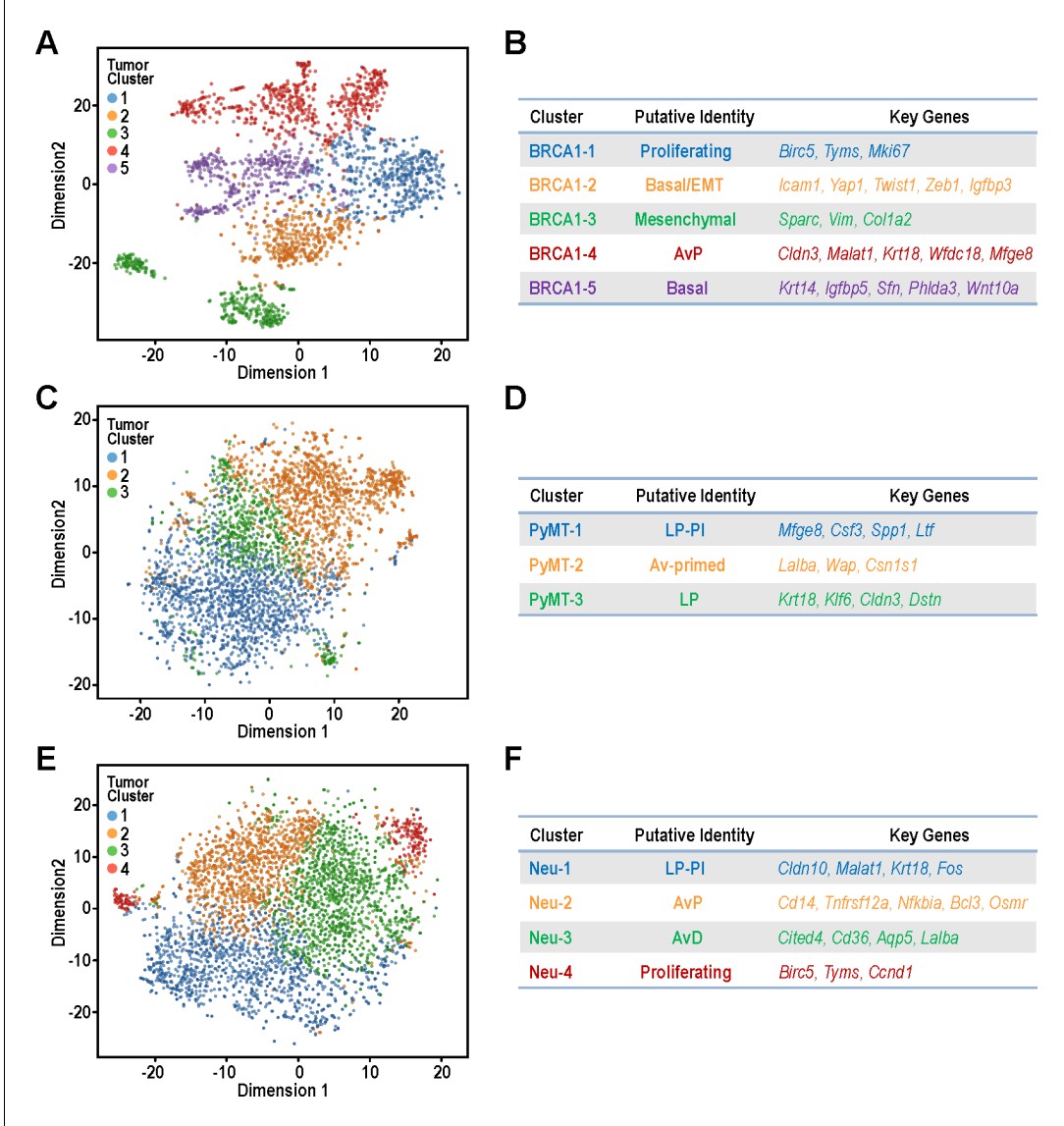

**Figure 4.** Identification of tumor sub-populations from distinct mouse models of breast cancer. (**A,C,E**) t-SNE plots for individual tumor samples (**A**) BRCA1-null tumor, (**C**) PyMT tumor and (**E**) Neu tumor, respectively. (**B,D,F**) Tables summarizing the sub-clusters, putative identities and key cluster defining genes; (**B**) BRCA1-null tumor, (**D**) PyMT tumor and (**F**) Neu tumor, respectively.

The online version of this article includes the following source data and figure supplement(s) for figure 4:

**Source data 1.** Differentially upregulated genes between sub-clusters in BRCA1-null tumors.
**Source data 2.** Differentially upregulated pathways (GO analysis) between sub-clusters in BRCA1-null tumors.
**Source data 3.** Differentially upregulated genes between sub-clusters in PyMT tumors.
**Source data 4.** Differentially upregulated pathways (GO analysis) between sub-clusters in PyMT tumors.
**Source data 5.** Differentially upregulated genes between sub-clusters in a 4T1 tumor.
**Source data 6.** Differentially upregulated genes between sub-clusters in Neu tumors.
**Source data 7.** Differentially upregulated pathways (GO analysis) between sub-clusters in Neu tumors.
**Figure supplement 1.** Even distribution of cells between respective tumor replicates in intra-tumoral analysis.
**Figure supplement 2.** Intra-tumoral heterogeneity within a 4T1 tumor.
**Figure supplement 3.** Cluster BRCA1-3 cells with mesenchymal characteristics were classified as tumor cells due to inferred presence of copy number variations (CNVs).
**Figure supplement 4.** Relapse free survival plots for breast cancer patients with gene expression signatures associated with BRCA1 clusters.

production related genes, this population had elevated expression of *Cd14*, *Tnfrsf12a*, *Nfkbia*, *Bcl3* and *Osmr*. The increased expression of *Cd14* in this Neu-2 cluster led us to the inference that these were AvP cells, since *Cd14* has been described as a marker for progenitor cells (*Bach et al., 2017*). Neu-1 cells did express *Cd14* along with increased LP-associated genes such as *Cldn10*, *Malat1*, and *Krt18*. Moreover, this cluster had lower expression of milk production genes, which indicated that these cells were most closely associated with C7:LP-PI cells. Neu-4 was a smaller cluster of proliferating cells characterized by elevated *Birc5*, *Tyms* and *Pcna*. The presence of Neu-4 cells in opposing ends of t-SNE dimension one suggested that there were distinct cell states with proliferative potential in this tumor model (*Figure 4E*).

Although multiple bioinformatics analyses of sc-RNAseq datasets from three tumor models did not reveal a universal BCSC population from the tumors examined, all these studies suggested the possibility that the less differentiated progenitor-like cells could act as lineage-specific TPCs in each particular tumor. We therefore took an experimental approach to test this idea using the Neu tumor due to its best-defined hierarchical structure from LPs (Neu-1) to AvPs (Neu-2) to AvDs (Neu-3). Both Neu-1 and Neu-2 clusters expressed higher levels of the luminal progenitor marker, *Cd14* (*Asselin-Labat et al., 2011*), along with other STAT3 target genes (*Bcl3*, *Osmr* [*Clarkson et al., 2004*]) and *Nfkbia* (*Figure 5A*). It is worth noting that the STAT3 pathway has been associated with acquisition of BCSC properties (*Wei et al., 2014*). We can also detect the presence of CD14[+] cells in a sub-population of Neu tumor cells by immuno-histochemistry, validating the scRNA-seq observations (*Figure 5B*). In order to evaluate the potential of CD14 as a cell surface marker for the isolation of putative TPCs in the Neu model, we generated a cell line (designated as N148 cells) from one of the Neu tumors that were analyzed in *Figures 1–4*. Through fluorescence assisted cell sorting (FACS), we isolated CD14[hi] and CD14[lo/-] populations from N148 cells (*Figure 5C*) and characterized these populations. Consistently, levels of *Cd14*, *Bcl3*, *Osmr* and *Nfkbia* were significantly increased in the CD14[hi] populations (*Figure 5D*). The increased expression of STAT3 target genes in the CD14[hi] population was also accompanied by elevated phospho-Stat3 levels (*Figure 5E*). When transplanted into syngeneic FvB mice at limiting dilutions, the CD14[hi] population also exhibited increased tumorigenic potential with an estimated TPC frequency of 1/278 compared to 1/2295 for CD14[lo/-] cells (Chi-squared test, p=0.00134) (*Figure 5F*). These results illustrated the TPC properties of luminal progenitor-like CD14[hi] cells within Neu tumors.

## Single-cell intrinsic molecular subtype assignment reveals breast cancer subtype heterogeneity within tumors

Although intrinsic molecular subtyping of breast tumors as a whole can be insightful in predicting prognosis and stratifying patients for therapy (*Sørlie et al., 2001*), these classification strategies do not account for the extensive heterogeneity at single-cell level. Since tumor cells corresponding to distinct cell states were observed within the respective tumors examined (*Figures 2–3*), we postulated that distinct intrinsic breast cancer subtypes would co-exist within such tumors (*Yeo and Guan, 2017*). To investigate this possibility, we utilized PAM50 and claudin-low signatures (*Prat et al., 2010*), which are routinely used for intrinsic molecular subtyping of breast tumors, to classify single mammary tumor cells using the Bioconductor package, genefu (*Gendoo et al., 2016*). Single cells were assigned intrinsic subtypes according to their respective correlations with claudin-low and PAM50 centroids (nearest centroid classifier). This approach has been utilized for single-cell analysis of human breast tumors (*Kim et al., 2018*). Interestingly, the Neu, PyMT and BRCA1-null tumors each comprised of cells which corresponded to several intrinsic breast cancer subtypes (*Figure 6A*). The BRCA1-null tumors had larger proportions of claudin-low and basal-like tumor cells (*Figure 6A*), whereas Neu and PyMT tumors contained more luminal A and normal-like subtype cells (*Figure 6A*). In general, this reflects the characteristics of the bulk tumors that were more basal-like (BRCA1-null) and luminal (Neu and PyMT) respectively. The presence of mesenchymal populations in BRCA1-null tumors (*Figures 3D* and *4A*) was also concordant with the assignment of claudin-low subtypes in this tumor type (*Figure 6A*). Altogether, the observation of multiple breast cancer subtypes within a tumor warrants further investigation into its potential clinical significance.

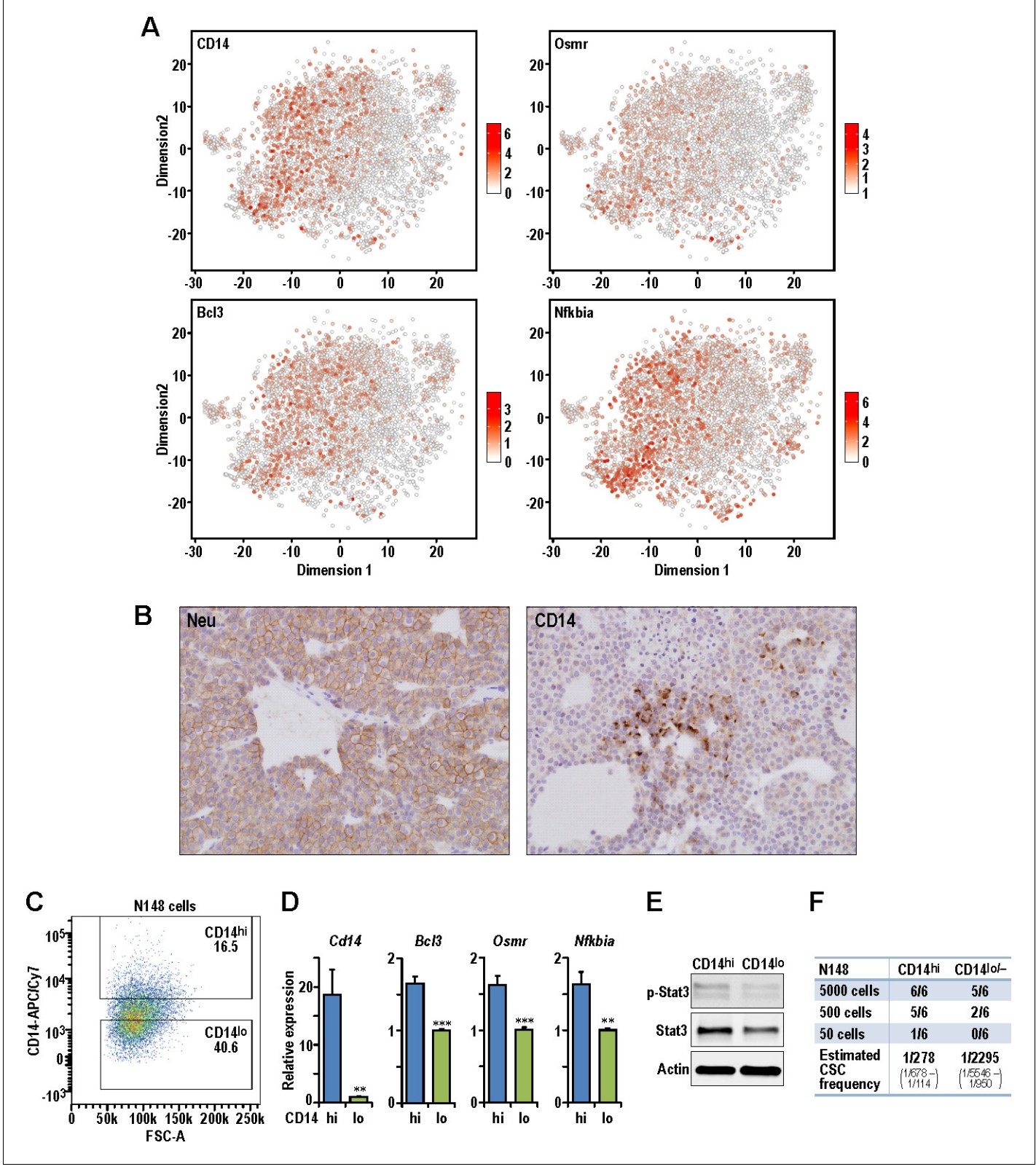

**Figure 5.** Isolation of CD14[hi] cells with increased tumorigenic potential in the Neu tumor. (**A**) t-SNE plots showing the expression levels of *Cd14, Bcl3, Osmr* and *Nfkbia* in the Neu tumor sub-clusters. (**B**) Representative immuno-histochemical staining of Neu and CD14 in a Neu tumor. (**C**) Dot plot showing the gating for isolation of CD14[hi] and CD14[lo/-] Neu tumor cell sub-populations. (**D**) Relative mRNA expression levels of *Cd14, Bcl3, Osmr* and *Nfkbia* between the sorted Neu tumor sub-populations. Bar charts show mean expression values and error bars represent standard error (n = 9,

*Figure 5 continued on next page*

*Figure 5 continued*

triplicates from three independent experiments, ** denotes p<0.01, *** denotes p<0.001). (**E**) Immuno-blots showing the levels of phospho-STAT3, STAT3 and ACTIN between the sorted Neu tumor sub-populations. (**F**) Limiting dilution transplant analysis of sorted Neu tumor sub-populations in syngeneic FvB mice. The appearance of tumors were recorded 4 weeks post-transplant and CSC frequencies were estimated using ELDA limiting dilution analysis (*Hu and Smyth, 2009*).

The online version of this article includes the following figure supplement(s) for figure 5:

**Figure supplement 1.** CD14 protein expression levels in mouse models and human breast tumors.

## Discussion

In this study, we analyzed the single-cell transcriptomes of 11639 mammary tumor cells from three prominent mouse models of breast cancer (BRCA1-null, PyMT, Neu). Interestingly, we found that tumor cells from these distinct models corresponded to minimally overlapping spectra across the mammary differentiation hierarchy (*Figure 6B*). In exact, BRCA1-null tumor cells corresponded to mesenchymal, basal and AvP lineages. On the other hand, PyMT tumor cells were associated with LPs of nulliparous and involution timepoints, whereas Neu tumor cells spanned the spectrum from LPs to AvPs to AvD cells. In a scenario with universal BCSCs, the universal BCSCs from different tumor models should cluster together in t-SNE analyses but this was not apparent from our analyses. However, the results presented here do not exclude the possibility of the existence of universal BCSCs, because they may exist at frequencies that were not detectable based on the cell numbers examined (<0.5%). The sorting of CD24$^+$ cells may also exclude cells that have undergone EMT but we utilized a less stringent CD24$^+$ gating strategy (*Figure 1—figure supplement 1*) that still enabled

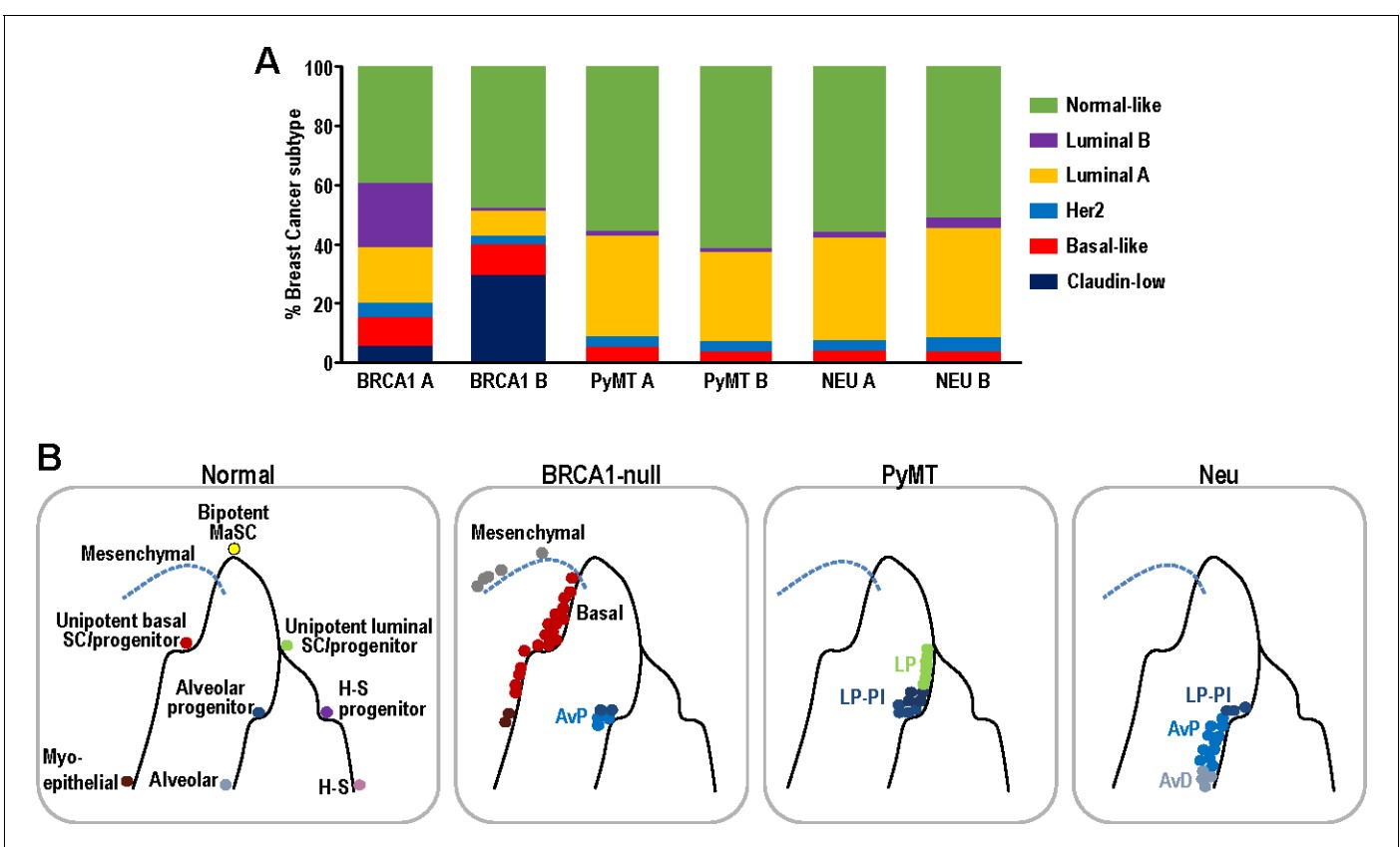

**Figure 6.** Single-cell intrinsic molecular subtype assignment of mammary tumor cells and model for the distribution of cell states within each mammary tumor. (**A**) Bar charts showing the % of cells representing claudin-low (navy blue), basal-like (orange), Her2 (blue), luminal A (yellow), luminal B (purple) and normal-like (green) breast cancer subtypes within BRCA1-null, PyMT and Neu tumors. (**B**) Schematic model summarizing the distribution of cell states for each of the tumors examined. Lines represent differentiation trajectories of MECs, while circles indicate cells.

the identification of mesenchymal populations, especially in the BRCA1-null tumors. Despite these potential limitations, our observations indicated that each of the tumors investigated were likely to contain lineage specific TPCs respectively. To illustrate this in the Neu tumor, we isolated LP-like and AvP-like cells through enrichment of CD14$^{hi}$ cells and showed that these cells exhibited increased tumorigenic potential in limiting dilution experiments, which fits the operational definition of TPCs (*Figure 5*). These CD14$^{hi}$ cells in the Neu tumors could be highly enriched for parity induced-MECs, that have been previously described to be BCSCs in this model (*Asselin-Labat et al., 2011*; *Henry et al., 2004*). In light of recent lineage-tracing studies showing that the basal, luminal ER$^+$ and luminal ER$^-$ MEC populations can respectively self-sustain (*Van Keymeulen et al., 2017*; *Van Keymeulen et al., 2011*; *Wang et al., 2017*), it is not too surprising if distinct TPCs contribute to the propagation of particular tumor subtypes/lineages.

We anticipate that tumors spanning a similar differentiation spectra (tumors with LP, AvP and AvD cells) will contain identical TPCs that can be isolated by the surrogate marker CD14. However, the illumination of distinct hierarchical patterns in the varying tumor types indicates that this marker will not be universally applicable to all tumor types. As an example, all the tumor cells within the PyMT model correspond to the LP cell state and in such a scenario CD14 is not differentially expressed between the tumor cell clusters (*Figure 4—figure supplement 1E*). Accordingly, from our previous studies (*Yeo et al., 2016*), TPCs in PyMT-driven tumors can be demarcated by higher expression of the more specific, primitive LP marker, *Aldh1a3* (*Eirew et al., 2012*). As for the BRCA1-null model (*Figure 4—figure supplement 1D*), it is likely that CD14 will demarcate LP cells that are of the pre-malignant stages before these cells trans-differentiate toward a more basal/mesenchymal cell state (*Wang et al., 2019*) and this will be an interesting prospect to be addressed in our future work. It is possible that Cluster BRCA1-3 cells (*Figure 4B*) were also trans-differentiated mesenchymal tumor cells because they were mutually exclusive from cells that had normal copy number variation (CNV) predictions (*Figure 4—figure supplement 3A*). Cells from cluster BRCA1-3 were classified as tumor because they were predicted to harbor chromosomal amplification or deletion events (*Figure 4—figure supplement 3B*). In order to examine whether there were any potential associations between the genes expressed by BRCA1 clusters with breast cancer patient survival, we used the aggregated mean expression of the top 50 differentially expressed genes for each BRCA1 cluster to analyze potential associations between survival of patients with high or low gene signatures for each cluster, respectively, utilizing KM-plotter (*Györffy et al., 2010*). Interestingly, this analysis revealed that patients expressing higher levels of the genes differentially expressed by clusters BRCA1-1 (Proliferating) and BRCA1-4 (Alveolar Progenitor-like) respectively (*Figure 4B*), were associated with poorer relapse-free survival (*Figure 4—figure supplement 4*). While the association between proliferation (cluster BRCA1-1) and poor prognosis may not be too surprising (*van Diest et al., 2004*), the observation that alveolar progenitor-like (cluster BRCA1-4) gene expression was associated with poorer survival is interesting and worthy of further examination in future work. Since CD14 protein expression can be detected in a subset of BRCA1-null mouse tumor cells (*Figure 5—figure supplement 1A*), as well as human breast cancers (*Figure 5—figure supplement 1B*), further work will be necessary to validate its potential significance in human breast cancer. In general, our finding of distinct patterns of cell state heterogeneity accentuates the need to specifically select and utilize markers to isolate relevant populations based on the respective tumor hierarchies, instead of universally employing surrogate TPC markers across all tumor types.

Importantly, our proposed model incorporates the elaborate intricacy of the mammary differentiation hierarchy, which is a conceptual leap from the focus on just universal BCSCs that reside at the apex of the tumor hierarchy. As in the case of the Neu tumor (*Figure 5*), progenitor-like cells and other intermediary states within the tumor hierarchy could also be relevant. Even the more differentiated cell states may have a role in supporting tumor growth through a paracrine manner and similar mechanisms of cooperativity between tumor sub-populations have been documented (*Cleary et al., 2014*; *Marusyk et al., 2014*). The clinical implications of our proposed model are that we cannot assume the presence of universal BCSCs in every tumor and prescribe universal BCSC targeting drugs without considering the specific stem/progenitor TPCs that may be driving them. The composition (hierarchical heterogeneity) of breast tumors may need to be accounted for to determine the combination of therapeutic agents that can effectively eliminate all the tumor sub-populations with discrete susceptibilities (*Yeo and Guan, 2017*).

With regard to that, intrinsic molecular subtyping of breast cancers can be associated with particular tumor differentiation cell states (*Visvader and Clevers, 2016*) and single-cell intrinsic molecular subtyping (*Chung et al., 2017*; *Yeo and Guan, 2017*) could potentially play a role in illuminating the spectrum of hierarchical heterogeneity within tumors. As we and others have shown (*Figure 6A*), multiple breast cancer subtypes can co-exist within a tumor (*Kim et al., 2018*) and there also appears to be some concordance between the tumor cell states and intrinsic molecular subtypes that were present. Even then, stratification strategies for single-cell classification will require further refinement because of the influence from cell-cycle-related genes. It would also be interesting to determine if tumor subtype composition at the single-cell level have prognostic value in breast cancer patients. Altogether, diagnosing the intra-tumoral breast cancer subtype heterogeneity may provide an indication of the hierarchical heterogeneity within tumors and guide the selection of combinatorial therapeutic regimens.

In summary, the data that was generated in this study provides insights into the degree of intra-tumoral heterogeneity within several mouse models of breast cancer. Although we have only analyzed these tumors under 'steady state' conditions, the single-cell data can serve as an important reference point for future studies which perturb the dynamics of the sub-populations. It would be interesting to investigate the changes in sub-populations after treatment, genetic manipulation, in secondary tumor sites and the degree of plasticity between these populations. In light of a hierarchical model for metastasis (*Lawson et al., 2015*), it is possible that certain sub-populations from these tumor models would be enriched in circulation, along the metastatic cascade. Although we did observe a more mesenchymal sub-population in BRCA1-null tumors and the process of EMT has been shown to confer stem-like properties (*Chaffer et al., 2013*), it would be interesting to identify other potential factors that may regulate tumor cell plasticity in some of the other tumor sub-populations (*Dravis et al., 2018*). Crucially, our results unraveled the distinct spectrum of differentiation states within mammary tumors and indicate that these tumors contain lineage-specific stem/progenitor-like TPCs. This emphasizes the need to account for the distinct spectrum of differentiation states within breast tumors to prevent therapeutic resistance.

# Materials and methods

## Key resources table

| Reagent type (species) or resource | Designation | Source or reference | Identifiers | Additional information |
|---|---|---|---|---|
| Genetic reagent (*Mus musculus*) | FVB/NJ | Jackson Laboratory | 001800 | RRID:IMSR_JAX:001800 |
| Genetic reagent (*Mus musculus*) | BALB/cJ | Jackson Laboratory | 000651 | RRID:IMSR_JAX:000651 |
| Genetic reagent (*Mus musculus*) | FVB/N-Tg(MMTV-neu)202Mul/J | Jackson Laboratory | 002376 | RRID:IMSR_JAX:002376 |
| Genetic reagent (*Mus musculus*) | FVB/N-Tg(MMTV-PyVT)634Mul/J | Jackson Laboratory | 002374 | RRID:IMSR_JAX:002374 |
| Genetic reagent (*Mus musculus*) | FVB/N-Brca1$^{f/f}$ Trp53$^{f/f}$ Tg(Krt14-Cre) | *Liu et al., 2007* | | Jos Jonkers RRID:MGI:3762188 |
| Cell line (*Mus musculus*) | N148 | This paper | | Primary cell line derived from Neu tumor. Description can be found under the 'Limiting dilution transplantation' section in Materials and methods |
| Cell line (*Mus musculus*) | FF99WT | This paper | | Primary cell line derived from PyMT tumor. Description can be found under the 'Animals' section in Materials and methods |
| Cell line (*Mus musculus*) | 4T1 | ATCC | CRL-2539 | RRID:CVCL_0125 |

*Continued on next page*

*Continued*

| Reagent type (species) or resource | Designation | Source or reference | Identifiers | Additional information |
|---|---|---|---|---|
| Antibody | Anti-CD14 (Rabbit polyclonal) | Invitrogen | PA5-78957 | IHC (1:200) RRID:AB_2746073 |
| Antibody | Anti-ErbB2 (Rabbit monoclonal) | Cell Signaling Technology | 2165 | IHC (1:200) RRID:AB_10692490 |
| Antibody | Anti-Mki67 (Rabbit monoclonal) | Spring Bioscience | M3062 | IHC (1:200) RRID:AB_11219741 |
| Antibody | Anti-Irf1 (Rabbit monoclonal) | Cell Signaling Technology | 8478 | IHC (1:200) RRID:AB_10949108 |
| Antibody | Anti-p-Stat3 (Rabbit monoclonal) | Cell Signaling Technology | 9145 | WB (1:1000) RRID:AB_2491009 |
| Antibody | Anti-Stat3 (Mouse monoclonal) | Cell Signaling Technology | 9139 | WB (1:1000) RRID:AB_331757 |
| Antibody | Anti-beta Actin (Mouse monoclonal) | Sigma | A5441 | WB (1:1000) RRID:AB_476744 |
| Antibody | PE Anti-CD24 (Rat monoclonal) | BD Biosciences | 553262 | Flow (1:200) RRID:AB_394741 |
| Antibody | APC Anti-CD31 (Rat monoclonal) | Biolegend | 102410 | Flow (1:500) RRID:AB_312905 |
| Antibody | APC Anti-TER119 (Rat monoclonal) | Biolegend | 116212 | Flow (1:500) RRID:AB_313713 |
| Antibody | APC Anti-CD45 (Rat monoclonal) | Biolegend | 103112 | Flow (1:500) RRID:AB_312977 |
| Antibody | APC/Cy7 Anti-CD14 (Rat monoclonal) | Biolegend | 123317 | Flow (1:200) RRID:AB_10900813 |
| Sequence-based reagent | CD14 F | IDT | qPCR primer | ACCGACCATGGAGCGTGTG |
| Sequence-based reagent | CD14 R | IDT | qPCR primer | GCCGTACAATTCCACATCTGC |
| Sequence-based reagent | BCL3 F | IDT | qPCR primer | CCGGAGGCCCTTTACTACCA |
| Sequence-based reagent | BCL3 R | IDT | qPCR primer | GGAGTAGGGGTGAGTAGGCAG |
| Sequence-based reagent | OSMR F | IDT | qPCR primer | CATCCCGAAGCGAAGTCTTGG |
| Sequence-based reagent | OSMR R | IDT | qPCR primer | GGCTGGGACAGTCCATTCTAAA |
| Sequence-based reagent | NFKBIA F | IDT | qPCR primer | TGAAGGACGAGGAGTACGAGC |
| Sequence-based reagent | NFKBIA R | IDT | qPCR primer | TTCGTGGATGATTGCCAAGTG |
| Commercial assay or kit | 10x Chromium single-cell kit | 10x Genomics | | V2 chemistry |
| Software, algorithm | Cell Ranger | 10x Genomics | | |
| Software, algorithm | Single-cell RNA sequencing analysis | This paper | | https://github.com/ZhuXiaoting/BreastCancer_SingleCell (copy archived at https://github.com/elifesciences-publications/BreastCancer_SingleCell) |

## Animals

All experimental procedures were carried out according to protocols approved by the Institutional Animal Care and Use Committee at University of Cincinnati under protocol number 13-10-08-02. Mice were housed and handled according to local, state and federal regulations. Brca1$^{F/F}$ P53$^{F/F}$

K14-Cre mice that have been previously described (*Liu et al., 2007*) were a kind gift from Dr. Jos Jonkers. MMTV-PyMT (*Guy et al., 1992a*) and MMTV-Neu (*Guy et al., 1992b*) strains were originally procured from Jackson Laboratory and were crossed with FIP200$^{F/F}$ mice as described previously (*Wei et al., 2011*). FF99WT cells were derived from an autochthonous MMTV-PyMT tumor with FIP200 floxed alleles (*Wei et al., 2011*) (essentially wild-type without Cre recombinase). FF99WT cells were used to generate the PyMT transplant tumors that were analyzed by scRNA-seq. For transplantation experiments, cells were prepared in DMEM:Matrigel at a 1:1 ratio and the required number of cells (4T1 = 10,000 cells; FF99WT = 1,000,000 cells) were injected in a 50 µl volume orthotopically into the fourth inguinal mammary fat pads. Wild-type FVB/NJ mice or BALB/C female mice (6–8 weeks old) were obtained from Jackson Laboratory for syngeneic transplantation of MMTV-PyMT tumor cells and 4T1 cells, respectively. 4T1 cells were obtained from ATCC and cultured in DMEM supplemented with 10% FBS. 4T1 cells were transduced with pLenti-GFP (LTV-400, Cell Biolabs) before transplantation experiments. All cell lines were routinely tested for mycoplasma monthly and were negative for contamination. The resulting tumors from spontaneous models (BRCA1-null and MMTV-Neu) along with transplant models (PyMT and 4T1) were harvested when tumor sizes were about 1000 mm$^3$ and isolated as described below.

## Tumor dissociation into single-cell suspension

Dissected primary tumors were cut and minced and then incubated in DMEM/F12 supplemented with 5% FBS, 1x antibiotic-antimycotics, gentamycin (20 µg/mL) (all from Invitrogen), collagenase (300 units/mL), and hyaluronidase (100 units/mL) (Worthington Biomedical) for 2 hr at 37°C. They were then centrifuged at 400 g for 5 min, and the pellets were washed with PBS, and suspended in 0.05% trypsin/0.025% EDTA (Invitrogen) before incubated for 5 min at 37°C. An equal volume of DMEM/F12 containing 5% FBS and DNase I (20 units/ml, Roche) was then added to stop the digestion. After filtering through a 40 µm nylon mesh (BD Biosciences), the cells were collected by centrifugation at 400 g for 5 min and re-suspended in ACK lysis buffer (Invitrogen) to lyse red blood cells.

## Flow cytometry and cell sorting

Cells were suspended in flow buffer (HBSS with 0.1% FBS and DNAse I) for analysis and sorting. Tumor single-cell suspensions were incubated with CD24-PE (553262, BD Biosciences), the endothelial cell marker, CD31-APC (102410, Biolegend), the leukocyte marker, CD45-APC (103112, Biolegend), and the erythrocyte marker, Ter119-APC (116212, Biolegend) according to manufacturer's instructions for 20 min at 4°C. Following that, cells were rinsed and resuspended in flow buffer before sorting or analysis by FACSAria or FACSCanto instruments (BD Biosciences). The gating strategy for tumor cell sorting was shown in *Figure 1—figure supplement 1B-C*. For isolation of CD14$^+$ cells, CD14-APC/Cy7 (123317, Biolegend) antibody was used and the gating strategy shown in *Figure 5C*.

## Library preparation and sequencing

Single-cell cDNA libraries were prepared using the 10x Chromium single-cell kit (3' gene expression platform version two chemistry) according to manufacturer's instructions for each tumor on separate chips. The aim was to capture about 2000 cells for each biological replicate and include two independent biological replicates for BRCA1-null, PyMT and Neu tumor models, respectively. Tumor libraries (BRCA1-null, PyMT and Neu) were then pooled and sequenced in a flow cell across two lanes using an Illumina HiSeq 2500 sequencer with the following parameters; read 1: 27 cycles; i7 index: eight cycles, i5 index: 0 cycles; read 2: 147 cycles. Independent biological replicates of tumor libraries were pooled and sequenced in a second batch. Library preparation and DNA sequencing were carried out by the CCHMC Gene Expression Core and CCHMC DNA Sequencing Core, respectively.

## RNA-sequencing data processing and quality control

The 10x Genomics workflow was followed to process the raw sequencing data. The Cell Ranger Single-Cell Software Suite was used for demultiplexing, barcode assignment and UMI quantification. The pre-built annotation package of mm10 reference genome was obtained from the 10x Genomics website and used as reference for read alignment by STAR. To combine data from seven sample

libraries, the outputs of each sample were aggregated using 'cellranger aggr' with –normalise set to 'none'. All together 13,745 cells were identified from four samples (1903 in 4T1, 3102 in BRCA1-null, 4173 in PyMT, 4567 in Neu).

For cell quality control, we excluded cells in which the number of genes detected and total number of unique molecular identifiers (UMIs) were three median absolute deviations (MAD) below the median of each sample, as well as total less than 1000 molecules or less than 500 genes detected. In addition, all cells with 20% or more of UMIs mapping to mitochondrial genes were defined as non-viable or apoptotic and removed from the analysis. We also avoided the effect of index swapping (non-unique barcodes) by excluding cells with barcodes that appeared in more than one sample. After quality control, there were a total of 11639 cells (1656 in 4T1, 2284 in BRCA1-null, 3545 in PyMT and 4154 in Neu). Pearson correlation test on the mean gene expression profiles showed high reproducibility between two biological replicates (*Figure 1—figure supplement 3*). To classify the cells into cell cycle phases, the function 'cyclone' from 'scran' package was applied using a pre-trained set of marker pairs for mouse data, described by *Buettner et al., 2015*; *Scialdone et al., 2015*. A score for each phase was assigned for each cell corresponding to the probability that the cell is in that phase, and cells were classified based on the phase score (*Figure 1—figure supplement 4*). In order to determine the percentage of normal mammary epithelial cells in the sample, copy number variation (CNV) calling was performed using the R package; CONICSmat (*Müller et al., 2018*). Briefly, the mouse chromosome region coordinates were obtained from mm10. The genes expressed in less than five cells were filtered out. The noisy regions of CNVs were filtered based on the results of the likelihood ratio test (LRT adjusted p-value<0.01) and BIC for each region (BIC difference >2000). By thresholding on the refined posterior probabilities at 0.8, the matrix is converted to binary where one represents the presence of CNV (tumor) and 0 the absence (normal).

## Clustering and differential expression analysis

The Bioconductor package 'scran' was used for the single cell RNAseq data processing. Three tumor types of the same genetic background were analyzed together and 4T1 tumor data were analyzed individually to exclude any potential confounding effects caused by differences in genetic background. Specifically, for three tumor types (BRCA1-null, PyMT and Neu), the gene expression counts in each cell were firstly scaled by size factors estimated by the 'computeSumFactors' function with default parameters in order to normalize the cell-specific bias. The normalized counts were log-transformed for downstream applications. For dimension reduction, the highly variable genes (HVGs) that drive biological heterogeneity were identified by modeling per gene variance and decomposing the total variance of each gene into its biological and technical components. Genes with the highest biological variant components were then examined and genes driven by technical noise were removed. Specifically, a mean-variance trend was fitted to the normalized log-expression values with 'trendVar' function to exhibit technical noise, and this value was subtracted from total variance to obtain biological components for each gene. The genes with biological variance $\geq 0$ and Benjamini-Hochberg adjusted p-values<0.1were selected as HVGs. Altogether 3178 HVGs were identified and used as features for PCA and top 50 PCs were used for t-SNE projection. The t-SNE embedding was computed using the 'Rtsne' package with default setting and perplexity set to 50.

The cells were clustered by a shared-nearest neighbor graph (SNN graph)-based method with default parameters. A shared nearest neighbor graph was first built with the expression data of the cells and the walktrap algorithm was used to identify clusters. The marker genes were identified using the 'findMarkers' function in 'scran' by testing for differential expression between clusters to identify the top upregulated genes in each cluster. The identified markers that ranked top 10 across all comparisons were selected to visualize by heat-map. (*Figure 1C*).

## Superimposing tumor dataset with normal mammary cell dataset

To characterize the hierarchical composition of each tumor model (*Figure 2*), we overlaid the cells with mouse epithelial cells by Bach et al. 12000 cells were randomly sampled from the published dataset (GES106273). The QC was conducted following the original processes and 11637 cells were left, consistent with the sample size of our data. The integration of normal and tumor data was implemented by Seurat v3, in order to identify common cell types and enable comparative analysis.

Specifically, Seurat objects were built individually for the tumor dataset and reference normal dataset and top 2000 variant genes in each object were identify as variable features. The cell pairwise correspondences between single cells between datasets, called 'anchors', were identified using 'FindIntegrationAnchors' by taking the Seurat objects list of tumor and normal using 20 dimensions, which jointly reduced the dimensionality of both datasets using diagonalized Canonical Component Analysis and L2-normalization. 5916 anchors were identified and used to integrate the two datasets by transforming the datasets into a shared space regardless of technical and biological difference.

To analyze the integrated dataset, PCA was applied for dimensionality reduction and t-SNE was used for visualization following the standard workflow. Clustering was also conducted by a shared nearest neighbor (SNN) modularity optimization-based clustering algorithm using default parameters (*Figure 3A–B*). The up-regulated gene markers of each cluster were identified using function 'FindAllMarkers' with log fold change threshold set as 1, and the markers ranked top five were selected to visualize by heatmap (*Figure 3C*). The cells that were assigned to each cluster were also annotated by their tumor origins (BRCA1-null, PyMT or Neu) or original clustering identities from Bach.et al. (*Figure 3D*).

## Diffusion maps, cell state trajectory and pseudo-time analyses

In order to explore the lineage profile of the different tumors and infer the differential trajectory, diffusion maps were built using 'Monocle3' with standard setting (*Trapnell et al., 2014*; *Figure 2—figure supplement 1*).

## Intra-tumoral heterogeneity analysis

To identify the intra-tumoral heterogeneity, four tumor samples were also analyzed individually using 'scran' following the standard workflow of normalization, dimension reduction and clustering. For three tumors with two replicates (BRCA1-null, PyMT or Neu), a batch correction process using function 'fastMNN' in 'scran' was applied on the two replicate samples, and the corrected values in low dimensional subspace were used for downstream analysis of clustering by SNN-graph clustering and visualization by t-SNE. The number of clusters (BRCA1-null = 5, PyMT = 3, Neu = 4 and 4T1 = 3) were selected based on differential expression of genes defining cell states. Cluster numbers were adjusted if a cluster did not express genes that could be correlated with a particular cell state or biological process. This justification provides a meaningful narrative for the clusters in each tumor type.

## Gene set and pathway analysis

The gene set enrichment analysis based on gene ontology (GO) terms was conducted to characterize the signature gene sets representing each cluster in each sample (*Figures 4–5*). The top 50 significantly upregulated genes in each cluster were compared to all the genes tested for differential expression in each tumor sample using 'topGO' with default setting.

## Intrinsic breast cancer molecular subtype assignment

To characterize the intrinsic molecular subtype of single cells in each sample (*Figure 6*), we compared the mouse single-cell expression data to human breast cancer molecular subtype assignment. Firstly, the genes that are one-to-one homologues between human and mouse were mapped according to Homologene (ftp://ftp.ncbi.nlm.nih.gov/pub/HomoloGene/build68/homologene.data). The symbol of mouse genes were mapped to human gene symbols and 10,855 out of 12438 genes were uniquely mapped. The 'genefu' package was used to assign the cell type of each single cell by PAM50 and claudin-low subtype assignment (*Gendoo et al., 2016*).

## Immunohistochemistry and immunoblotting

Formalin-fixed paraffin-embedded tumors were sectioned (5 µM) and stained for respective antigens as described previously (*Wei et al., 2011*). Slides were heated in citrate buffer in a pressure cooker for antigen retrieval. Antibodies used for immuno-staining included CD14 (PA5-78957, Invitrogen), Erbb2 (CST2165, Cell Signaling), Irf1 (CST8478, Cell Signaling) and Mki67 (M3062, Spring Bioscience). Immunoblotting experiments were carried out as described previously (*Yeo et al., 2016*). Antibodies used for immunoblotting were p-STAT3 (CST9145, Cell Signaling), STAT3 (CST9139, Cell Signaling) and beta-Actin (A5441, Sigma Aldrich).

## Quantitative PCR

Total RNA was isolated from cells using RNAeasy kit (Qiagen) according to manufacturer's instructions. Equal amounts of RNA were then reverse-transcribed using SuperScript III first-strand synthesis kit (Invitrogen) using random hexamers as primers. cDNA samples were then subjected to qRT-PCR analysis with SYBR Green in a BioRad CFXConnect thermo-cycler. Primers used were; CD14 F: 5'-ACCGACCATGGAGCGTGTG-3', CD14 R: 5'-GCCGTACAATTCCACATCTGC-3', BCL3 F: 5'-CCGGAGGCCCTTTACTACCA-3', BCL3 R: 5'-GGAGTAGGGGTGAGTAGGCAG-3', OSMR F: 5'-CATCCCGAAGCGAAGTCTTGG-3', OSMR R: 5'- GGCTGGGACAGTCCATTCTAAA-3', NFKBIA F: 5'-TGAAGGACGAGGAGTACGAGC-3', NFKBIA R: 5'- TTCGTGGATGATTGCCAAGTG-3'.

## Limiting dilution transplantation

N148 cells were derived from the autochthonous MMTV-Neu tumor with FIP200 floxed alleles (essentially wild-type without Cre recombinase) that was analyzed by sc-RNAseq (Neu_A). For limiting dilution transplants, mice were monitored for 1 month for the formation of tumors.

## Statistical analysis

For bar charts, data were presented as means and error bars indicated standard error of the mean (SEM). In *Figure 5D*, statistical significance was evaluated by unpaired t-test using $p < 0.05$ as indicative of statistical significance. For *Figure 5F*, TPC frequencies and statistical differences between groups for limiting dilution transplants were performed using ELDA as described previously (*Hu and Smyth, 2009*).

## Availability of data and materials

The authors declare that all data supporting the findings of this study are available within the article and its supplementary information files or from the corresponding author upon reasonable request. The single-cell RNA-seq data have been deposited in the GEO database under accession code GSE123366. Computational analyses were performed in R (version 3.6.0) using standard functions unless otherwise stated. Code is available online at https://github.com/ZhuXiaoting/BreastCancer_SingleCell (copy archived at https://github.com/elifesciences-publications/BreastCancer_SingleCell).

## Acknowledgements

This work was supported by NIH grants R01-CA211066, R01-HL073394 and R01-NS094144 to JLG, and XZ is partially supported by an NIH grant R01-HL111829 to LJL. We are extremely grateful to Jos Jonkers (Netherlands Cancer Institute) for providing *Brca1*$^{F/F}$; *Trp*$^{F/F}$; K14-Cre mice and Xiaoting Zhang for MMTV-Neu mice. We thank our colleagues Nathan Salomonis, Tom Cunningham, Susan Waltz, Phillip Dexheimer, Maria Czyzyk-Krezska, Chenran Wang, Xin Tang, Michael Haas and Ritama Paul for critical appraisal and suggestions in the preparation of this manuscript. We appreciate the help from Glenn Doerman for graphics support.

## Additional information

### Funding

| Funder | Grant reference number | Author |
| --- | --- | --- |
| National Institutes of Health | R01-CA211066 | Jun-Lin Guan |
| National Institutes of Health | R01-HL111829 | Long Jason Lu |
| National Institutes of Health | R01-NS094144 | Jun-Lin Guan |
| National Institutes of Health | R01-HL073394 | Jun-Lin Guan |

The funders had no role in study design, data collection and interpretation, or the decision to submit the work for publication.

## Author contributions
Syn Kok Yeo, Conceptualization, Formal analysis, Investigation, Methodology, Writing - original draft, Writing - review and editing; Xiaoting Zhu, Data curation, Software, Formal analysis, Investigation, Visualization, Methodology, Writing - review and editing; Takako Okamoto, Mingang Hao, Formal analysis, Investigation; Cailian Wang, Formal analysis, Investigation, Visualization; Peixin Lu, Formal analysis; Long Jason Lu, Resources, Formal analysis, Supervision, Funding acquisition, Investigation, Methodology, Project administration, Writing - review and editing; Jun-Lin Guan, Conceptualization, Resources, Supervision, Funding acquisition, Writing - original draft, Project administration, Writing - review and editing

## Author ORCIDs
Syn Kok Yeo (iD) https://orcid.org/0000-0002-4333-2510
Jun-Lin Guan (iD) https://orcid.org/0000-0001-8720-8338

## Ethics
Animal experimentation: All experimental procedures were carried out according to protocols approved by the Institutional Animal Care and Use Committee at University of Cincinnati under protocol number 13-10-08-02. Mice were housed and handled according to local, state and federal regulations.

## Decision letter and Author response
Decision letter https://doi.org/10.7554/eLife.58810.sa1
Author response https://doi.org/10.7554/eLife.58810.sa2

# Additional files

## Supplementary files
• Transparent reporting form

## Data availability
The authors declare that all data supporting the findings of this study are available within the article and its supplementary information files or from the corresponding author upon reasonable request. The single-cell RNA-seq data have been deposited in the GEO database under accession code GSE123366. Computational analyses were performed in R (version 3.6.0) using standard functions unless otherwise stated. Code is available online at https://github.com/ZhuXiaoting/BreastCancer_SingleCell (copy archived at https://github.com/elifesciences-publications/BreastCancer_SingleCell).

The following dataset was generated:

| Author(s) | Year | Dataset title | Dataset URL | Database and Identifier |
|---|---|---|---|---|
| Yeo SK, Zhu X, Lu LJ, Guan JL | 2018 | Single-cell transcriptomic analysis of mammary tumors reveals distinct patterns of hierarchical and subtype heterogeneity | https://www.ncbi.nlm.nih.gov/geo/query/acc.cgi?acc=GSE123366 | NCBI Gene Expression Omnibus, GEOGSE123366 |

The following previously published dataset was used:

| Author(s) | Year | Dataset title | Dataset URL | Database and Identifier |
|---|---|---|---|---|
| Bach K, Sara P | 2017 | Differentiation dynamics of mammary epithelial cells revealed by single-cell RNA-sequencing | https://www.ncbi.nlm.nih.gov/geo/query/acc.cgi?acc=GSE106273 | NCBI Gene Expression Omnibus, GEOGSE106273 |

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
