## [Decision Letter]

**Acceptance summary:**

The work presents highly interesting novel findings on distinct cellular states in mouse breast cancer models, which have minimal overlap between different oncogenic drivers. These insights stress the need for a deeper interrogation on the hierarchical composition of different breast cancer subtypes.

**Decision letter after peer review:**

Thank you for submitting your article "Single-cell transcriptomic analysis of mammary tumors reveal distinct patterns of hierarchical and subtype heterogeneity" for consideration by *eLife*. Your article has been reviewed by three peer reviewers, including Wilbert Zwart as the Reviewing Editor and Reviewer #1, and the evaluation has been overseen by Maureen Murphy as the Senior Editor. The following individuals involved in review of your submission have agreed to reveal their identity: Luca Magnani (Reviewer #2); Rachael Natrajan (Reviewer #3).

The reviewers have discussed the reviews with one another and the Reviewing Editor has drafted this decision to help you prepare a revised submission.

Summary:

In this work, the authors have analyzed >11K cells from three mouse models of breast cancer and have utilized previous knowledge of mammary development to explore the idea of a "normal-inspired hierarchy" within each tumour. Overall, the data are quite descriptive, with the exception of the re-validation of CD14 as a putative marker in basal like tumours. The final set of analysis are designed to identify heterogeneity within each tumor by classifying single cells into widely accepted human breast cancer subtypes. While the dataset is interesting, my opinion is that the manuscript falls a bit short from its initial question: are different cancer subtypes driven by a common stem cells which then differentiate toward different phenotypes? This was a hard question and I think the results here support quite clearly the most prevalent view which state that different tumor subtypes are likely originating in different cell of origin within the normal breast architecture.

Essential revisions:

1) There are some missing piece of data which I believe are required. The authors suggest that BRCA, Neu and PyMT recapitulate somehow Basal (TNBC), HER2 and Luminal cancer. It would be helpful if the authors could stain their samples for the classical marker used in the clinic: ER, PR and HER2. This is especially important for the interpretation of Figure 6, which is rather overstated at the moment (since PyMT and Neu are basically identical while only BRCA1B shows somehow distinct composition). BRCA1B and A have been profiled at very different levels (S2 E).

2) The hierarchy in Figure 6 is also over-speculative, since the authors themselves state that they could not really reconstruct it for BRCA1 and PyMT mices. This would have to be addressed. Also, what is the difference between BRCA1 and BRCA2 here? Isn't is strange that the MMTV-Neu model does not show increase in HER2 subtype on PAM50?

3) Could the authors explain their choices of gating for the CD14 transplantation experiment? Was it somehow informed by the single cell level of expression of CD14? Also, could the authors stain some Basal Human cancers for CD14 to check it's expression? Or do they believe this is a mouse-specific finding?

4) The authors state that "In summary, the data that was generated in this study provides a comprehensive level of insight into the degree of intra-tumoral heterogeneity". With about 2K cells per tumor model profiled at 2.5K genes per cell, I think this statement is inappropriate. A much bigger effort is required to achieve a comprehensive insight of heterogeneity (just an example, mapping at single cell level about 10K CD14 cells and CD14^-^ cells just for the BRCA model). Furthermore, the authors did not really attempt to perform more sophisticated corrections to exclude differences driven by cell cycle or other confounding factor.

5) The authors state "Levels of some alveolar differentiation genes such as Cited4 and *Tspan8* were marginally higher in Cluster 1." What do they mean by marginal? Is this not significant?

6) "On the other hand, BRCA1-3 cells expressed the highest levels of genes associated with fibroblasts (Figure 4B) and this would fit with our previous observations of a sub-population of BRCA1 null cells with mesenchymal characteristics (Figures 1-3)." Can the author's rule out that these cells are not fibroblasts?

7) Do the BRCA1 clusters identified have clinical relevance in human breast cancers?

8) It is not really clear from the title, nor from the Introduction, that the study is basically a basic science report: do different GEMM models of breast cancer have distinct or overlapping subpopulations of tumor cells? I'd recommend the authors change the title and Introduction accordingly, as especially the title now conveys a rather clinical message, that the study simply doesn't deliver.

9) The BRCA1 model is K14-CRE driven, while the other two are MMTV. Could this difference have an impact on the observed differences on cell populations? Now, the study is comparing three difference mouse models with different drivers, and states the tumors that develop are also different. But without a confirmation on a second independent mouse model representing the same subtype of breast cancer (but with a different genetic background, or a different promoter driving the phenotype), one cannot state whether the differences that are found, are due to the different tumor drivers, tumour subtypes or different models. I feel this is an important issue.

---

## [Author Response]

Summary:In this work, the authors have analyzed >11K cells from three mouse models of breast cancer and have utilized previous knowledge of mammary development to explore the idea of a "normal-inspired hierarchy" within each tumour. Overall, the data are quite descriptive, with the exception of the re-validation of CD14 as a putative marker in basal like tumours. The final set of analysis are designed to identify heterogeneity within each tumor by classifying single cells into widely accepted human breast cancer subtypes. While the dataset is interesting, my opinion is that the manuscript falls a bit short from its initial question: are different cancer subtypes driven by a common stem cells which then differentiate toward different phenotypes? This was a hard question and I think the results here support quite clearly the most prevalent view which state that different tumor subtypes are likely originating in different cell of origin within the normal breast architecture.

We thank the reviewers for the constructive comments. We would like to clarify that experimental revalidation of CD14 as a new putative CSC marker was carried out in Her2^+^ MMTV-Neu tumors.

We agree that our study does not touch upon the cell of origin or tumor initiating cells (TICs) in different breast cancer subtypes. We mean tumor propagating cells (TPCs) when we refer to BCSCs and not TICs. These confounding terms have been described in “Cell-of-Origin of Cancer versus Cancer Stem Cells: Assays and Interpretations” (Rycaj et al., Cancer Research, 2015). For that reason, we have used the term tumor propagating cell (TPC) when describing the results to differentiate them from TICs, after introducing the CSC concept initially.

Our emphasis was placed on tumor propagating cells (TPCs) in established tumors and our results identified distinct tumor hierarchies in each of the mouse models examined. We also agree that this observation is more in line with the prevalent view of distinct cell of origin for varying breast cancer subtypes.

Essential revisions:1) There are some missing piece of data which I believe are required. The authors suggest that BRCA, Neu and PyMT recapitulate somehow Basal (TNBC), HER2 and Luminal cancer. It would be helpful if the authors could stain their samples for the classical marker used in the clinic: ER, PR and HER2.

We thank the reviewer for the suggestion and have now included data to show levels of ER, PR and HER2 in the three tumor models analyzed (Figure 1—figure supplement 1A). From the additional data, the MMTV-Neu tumors examined were ER^-^/PR^-^/HER2^+^ while the MMTV-PyMT tumors were ER^‑^/PR^-^/HER2^lo^ (Figure 1—figure supplement 1A). On the other hand, BRCA1-null tumors were ER^-^/PR^-^/HER^-^ (Figure 1—figure supplement 1A) and this is consistent with Neu and PyMT tumors designated as luminal, whereas BRCA1-null tumors were designated as basal. These additional descriptions and Figure 1—figure supplement 1A have been added to the manuscript.

This is especially important for the interpretation of Figure 6, which is rather overstated at the moment (since PyMT and Neu are basically identical while only BRCA1B shows somehow distinct composition).

Both the MMTV-Neu and MMTV-PYMT tumors have a gene expression pattern that corresponds to luminal breast cancer (not Her2-enriched) (PMID: 17493263) and this would fit with their similar profiles in Figure 6. However, we realize that there are potential limitations of using PAM50 classification at the single-cell level, since this gene-list was originally developed for bulk tumors. Accordingly, we have stated in our Discussion that “Even then, stratification strategies for single-cell classification will require further refinement because of the influence from cell cycle related genes. It would also be interesting to determine if tumor subtype composition at the single-cell level have prognostic value in breast cancer patients.” to tone down our interpretations related to Figure 6A.

BRCA1B and A have been profiled at very different levels (S2 E).

We agree that the BRCA1 tumors were profiled at different levels but this was due to unexpected technical difficulties since we aimed to capture similar amount of cells for the BRCA1-null tumor replicates using the Chromium 10X system.

2) The hierarchy in Figure 6 is also over-speculative, since the authors themselves state that they could not really reconstruct it for BRCA1 and PyMT mices. This would have to be addressed.

The model shown in Figure 6 is an illustration to summarize the overlaps between tumor and normal mammary epithelial cell populations based on observations in Figures 1 to 3, so we do not think it is over-speculative. Although we are showing the cell states that are present in each tumor type, we did not specifically stipulate the TPCs for each model. To clarify further, we did not state that the hierarchies cannot be reconstructed for the BRCA1 and PyMT mice, but rather it was more difficult to infer TPCs from the hierarchies in these other models. To address this, we have now changed the statement to:

“They also provided interesting information for each cluster of cells within individual tumors, although the TPCs and putative markers to be used for isolation could not be deduced easily based on the current analysis so far, except for that in Neu cells discussed below.”

In PyMT mice, the reason was because the hierarchy was very shallow (illustrated by Figure 6A), since the tumor consisted largely of a luminal progenitor-like component (Figures 2B and 2F). However, we have previously shown that more primitive ALDH^+^ populations (primitive luminal progenitors) can be isolated amongst the bulk of luminal progenitor-like cells and the ALDH^+^ populations were more tumorigenic than bulk PyMT tumor cells (Yeo et al., 2016).

In the case of BRCA1-null tumors, the hierarchy is more complex with luminal progenitor-like, basal-like, hybrid alveolar-basal cells and mesenchymal populations (Figure 2C). As shown in Figure 4—figure supplement 1D and Figure 4A-B, it is likely that CD14 will demarcate luminal progenitor-like cells, which are found in early stage BRCA1-null tumors before these cells trans-differentiate towards more basal/mesenchymal cells (Wang et al., 2019). Thus, it is unclear whether CD14^+^ cells within this tumor type are the most tumorigenic (TPCs) or least differentiated. Hence, we will continue to identify markers that would allow us to isolate distinct basal, mesenchymal and luminal progenitor populations in this tumor model to compare their tumorigenic potentials as part of our future work. The combination of CD14 with a basal marker to potentially identify cells with hybrid alveolar-basal cells (presumably more stem-like and bipotent) in this tumor model is also an interesting prospect that we are looking into.

Also, what is the difference between BRCA1 and BRCA2 here?

The BRCA1 A and BRCA1 B are two independent tumor samples from separate mice. Since the BRCA1-null model is more heterogeneous (PMID: 22396490), due to genomic instability (loss of BRCA1 and p53), it makes sense that there is a wider range of variation for this model, as previously reported. Even so, the cell states that are present in both replicates are similar, albeit their proportions being different (Figure 4—figure supplement 1A and Figure 4A).

Isn't is strange that the MMTV-Neu model does not show increase in HER2 subtype on PAM50?

As mentioned in our response to point 1, the MMTV-Neu tumors have a gene expression pattern that corresponds to luminal breast cancer (PMID: 17493263). Thus, it may not be strange that the Neu model does not show higher numbers of cells corresponding to the Her2-enriched subtype. Furthermore, the PAM50 classification for the Her2-enriched subtype includes genes such as Grb7, which are frequently co-amplified with Her2 due to their proximity. Thus, co-amplification events are likely required along with Her2 over-expression to give rise to the Her2-enriched subtype (PMID: 12941816). Essentially, the MMTV-Neu model corresponds to the histological derived subtype of HER2^+^ but not the intrinsic subtype of HER2-enriched (PMID: 17493263).

3) Could the authors explain their choices of gating for the CD14 transplantation experiment? Was it somehow informed by the single cell level of expression of CD14?

The gating for CD14^+^ and CD14^-^ in Figure 5C was spaced out to minimize potential spillover of the two populations during the cell sorting process. However, it was not informed by the expression levels from the scRNAseq data.

Also, could the authors stain some Basal Human cancers for CD14 to check its expression? Or do they believe this is a mouse-specific finding?

As mentioned in our response to the summary above and point #2 (third paragraph), CD14 was defined as a putative marker for TPCs in the Her2^+^ Neu tumors and not basal-like tumors. Nonetheless, we expect to see CD14 staining in tumors of various subtypes, since BRCA1-deficient basal-like human cancers have also been shown to originate from luminal progenitor cells (PMID: 27322743, 21336547). This would also be reflective of the mRNA expression patterns of CD14 in BRCA1-null, PyMT and Neu tumors (Figures 4A, Figure 4—figure supplement 1D-E), where sub-populations of CD14 expressing cells were present in BRCA1-null and Neu tumors, whereas PyMT tumors exhibited more homogenous expression of CD14. We have now included Figure 5—figure supplement 1A to corroborate the CD14 mRNA data at the protein level by immuno-histochemistry.

In addition, we have also stained a panel of human breast cancer samples for CD14 (Figure 5—figure supplement 1B-C) and CD14 expression can be found across different histological subtypes without an association with a particular subtype in human breast cancer.

Overall, we agree that at present, the CD14 findings need to be confirmed in human breast cancer, hence we have changed the title of the manuscript to reflect that our study is focused on mouse models of breast cancer (Point #8).

4) The authors state that "In summary, the data that was generated in this study provides a comprehensive level of insight into the degree of intra-tumoral heterogeneity". With about 2K cells per tumor model profiled at 2.5K genes per cell, I think this statement is inappropriate. A much bigger effort is required to achieve a comprehensive insight of heterogeneity (just an example, mapping at single cell level about 10K CD14 cells and CD14^-^ cells just for the BRCA model). Furthermore, the authors did not really attempt to perform more sophisticated corrections to exclude differences driven by cell cycle or other confounding factor.

We have toned down the statement, based on the reviewers’ advice.

5) The authors state "Levels of some alveolar differentiation genes such as Cited4 and Tspan8 were marginally higher in Cluster 1." What do they mean by marginal? Is this not significant?

We used the term marginal because there was a 1.88 fold and 1.55 fold difference in the levels of Cited4 and Tspan8 respectively between Clusters 1 and 2. In contrast, the fold change between Clusters 1 and the other (3-6) clusters were much higher (Figure 1D). For both these genes, the p-values were 1.13 x 10^-104^ and 6.89 x 10^-93^ respectively (pairwise t-test between Clusters 1 and 2), which is statistically significant.

6) "On the other hand, BRCA1-3 cells expressed the highest levels of genes associated with fibroblasts (Figure 4B) and this would fit with our previous observations of a sub-population of BRCA1 null cells with mesenchymal characteristics (Figures 1-3)." Can the author's rule out that these cells are not fibroblasts?

The reviewer is right in pointing out that we cannot completely rule out that these are bona fide fibroblasts. However, we now provide additional data (Figure 4—figure supplement 3A) to show that Cluster BRCA1-3 cells are mutually exclusive from cells that had normal copy number variation (CNV) predictions. In line with this, cells from cluster BRCA1-3 were classified as tumor because they were predicted to harbor chromosomal amplification and/or deletion events (Figure 4—figure supplement 3B). This suggests that cells from this cluster may have undergone trans-differentiation.

In support of this view, recent tracing experiments in combination with single-cell RNA sequencing of BRCA1-null mouse mammary tumors has also revealed that trans-differentiation of tumor epithelial cells into mesenchymal populations does indeed occur upon BRCA1 deletion (Wang et al., 2019). This point has now been added to the Discussion section.

7) Do the BRCA1 clusters identified have clinical relevance in human breast cancers?

In order to examine whether there were any potential associations between the genes expressed by BRCA1 clusters with patient survival, we used the aggregated mean expression of the top 50 differentially expressed genes for each BRCA1 cluster to analyze potential associations between survival of patients with high or low gene signatures for each cluster respectively. This was carried out with KM-plot (PMID: 20020197). Interestingly, this analysis revealed that patients expressing higher levels of the genes differentially expressed by clusters BRCA1-1 (Proliferating) and BRCA1-4 (Alveolar Progenitor-like) respectively (Figure 4B), were associated with poorer relapse free survival (Figure 4—figure supplement 4). While the association between proliferation (cluster BRCA1-1) and poor prognosis may not be too surprising (van Diest et al., 2004), the observation that alveolar progenitor-like (cluster BRCA1-4) gene expression was associated with poorer survival is interesting and worthy of further examination in future work. This description has been added to the Discussion.

8) It is not really clear from the title, nor from the Introduction, that the study is basically a basic science report: do different GEMM models of breast cancer have distinct or overlapping subpopulations of tumor cells? I'd recommend the authors change the title and Introduction accordingly, as especially the title now conveys a rather clinical message, that the study simply doesn't deliver.

We would like to thank the reviewer for the suggestion and have edited the title as well as Introduction to better reflect that the manuscript is basic science oriented.

9) The BRCA1 model is K14-CRE driven, while the other two are MMTV. Could this difference have an impact on the observed differences on cell populations? Now, the study is comparing three difference mouse models with different drivers, and states the tumors that develop are also different. But without a confirmation on a second independent mouse model representing the same subtype of breast cancer (but with a different genetic background, or a different promoter driving the phenotype), one cannot state whether the differences that are found, are due to the different tumor drivers, tumour subtypes or different models. I feel this is an important issue.

We appreciate the point made by the reviewer and agree that the variables mentioned could contribute to the different cell states that were present in each tumor model. However, we did not state that these differences were specifically due to a particular variable such as driver oncogene or promoter and that was not the point we were trying to make.

Our main aim was to utilize these diverse mouse models to illustrate and exemplify a sampling of “mouse patients”. Accordingly, our findings showed that each of them had distinct patterns of cell state heterogeneity. This general observation itself is very important because one of the prevailing views regarding breast CSCs/TPCs is that they are universally the same entity (the stem-like cell on top of the mammary differentiation hierarchy), and the same entity is presumed to be present in all breast tumors. Our findings indicate otherwise as shown in Neu tumors, where CD14^+^ luminal progenitor-like cells (not stem-like) which occupy an intermediate state (not the apex) in the mammary differentiation hierarchy were shown to be TPCs. This would advocate for the need to account for potentially unique hierarchies in a particular mammary tumor rather than assuming universal breast CSCs.